# QUANTUM-INSPIRED STRUCTURE-AWARE DIFFUSION FOR EFFICIENT 3D MOLECULAR GENERATION

## ABSTRACT

The high computational cost of classical diffusion models can limit their use in large-scale 3D molecular generation for drug and material discovery. We introduce **S**tructure-aware **Q**uantum **Diff**usion (SQ-Diff), the first full quantum diffusion model for this task, designed to leverage potential quantum advantages in the Noisy Intermediate-Scale Quantum era. Structural priors (e.g., inter-atomic distances) are encoded into the initial quantum state via a novel state preparation procedure that yields a unified normalization scheme dependent only on the number of atoms. The denoising process is driven by a Quantum U-Net, a fully quantum architecture that combines learnable variational quantum circuits with parameter-free operators. Training is guided by these structural priors enforced through a graph-based objective function to maintain structural consistency. Experimentally, SQ-Diff generates valid and diverse 3D molecules and shows improved performance over existing quantum-based methods. While a gap in generation quality compared to leading classical models remains, our model matches the inference speed of the fastest classical approaches with only a few quantum parameters, setting a new benchmark for pure quantum generative models.

## 1 INTRODUCTION

The automated discovery of novel molecules with desired 3D structures is a key area of modern drug discovery and material science (Sanchez-Lengeling & Aspuru-Guzik, 2018). By generating valid and diverse molecules, computational methods can significantly accelerate the design-test-analyze cycle, reducing costs and exploring vast regions of the chemical space that are otherwise inaccessible. Among various deep generative models, such as Variational Autoencoders (VAEs) that learn a latent representation of the data (Kingma & Welling, 2013) and Generative Adversarial Networks (GANs) that use a discriminator to refine generated samples (Goodfellow et al., 2014), diffusion models (Ho et al., 2020; Song et al., 2021b) have recently emerged as a powerful paradigm for 3D molecular generation. Their performance stems from their formulation as score-based models that learn the gradient of the data log-likelihood, allowing them to capture fine-grained details of the underlying distribution. This has led to state-of-the-art performance in generating high-quality molecular geometries (Hoogeboom et al., 2022; Xu et al., 2023). However, despite their success, classical diffusion models face persistent challenges. The iterative denoising process is computationally intensive, leading to slow sampling speeds that can hinder high-throughput virtual screening (Wang et al., 2025). Furthermore, generative models can struggle to produce novel scaffolds beyond what is represented in their training data, a well-documented challenge in benchmarking de novo molecular design (Brown et al., 2019).

Quantum computing offers a fundamentally different paradigm for generative modeling, holding the potential to overcome some of these classical limitations (Biamonte et al., 2017; Smaldone et al., 2025). The principles of quantum mechanics, such as superposition and entanglement, provide access to an exponentially large Hilbert space. This vast state space is particularly well-suited for representing the intricate conformational and electronic states of molecules, which are themselves quantum systems. This allows quantum models to represent and explore complex, high-dimensional probability distributions with potentially greater efficiency and expressivity than their classical counterparts (Cerezo et al., 2021). Recent work has already demonstrated that hybrid quantum-classical models can generate novel, synthesizable molecules that are validated in the lab, showcasing the potential of this paradigm to impact real-world drug discovery campaigns (Ghazi Vakili et al., 2025).

Despite this promise, the application of Quantum Machine Learning (QML) to molecular generation is still nascent (Smaldone et al., 2025). Early efforts, such as quantum GANs with a hybrid generator (QGAN-HG) (Li et al., 2021) and scalable quantum generative autoencoders (SQ-VAE) (Li & Ghosh, 2022), relied on hybrid quantum-classical architectures, focusing on 2D molecular graphs. The more challenging task of direct 3D molecular generation in a pure quantum framework remains largely unexplored. While the recently proposed quantum circuit-based VAE for 3D molecular generation (QVAE-Mole) (Wu et al., 2024) represents a step forward by using a pure quantum VAE for 3D generation, limitations persist. These models often use state preparation procedures that do not fully embed the graphical topology, particularly inter-atomic distances. Failing to encode this geometric information can lead to structurally invalid outputs. Moreover, graph priors such as permutation invariance are not explicitly considered (Zhao et al., 2024). VAE-based approaches can also be prone to posterior collapse (Bowman et al., 2016). These limitations highlight the need for a more robust and structurally-aware quantum generative framework.

To address these challenges, we introduce, to our knowledge, the first pure quantum diffusion model for de novo 3D molecular generation. Our approach integrates molecular graph structure with a quantum diffusion process. We propose a novel state preparation procedure that encodes atom and bond information into the initial quantum state, creating a latent representation that reflects the molecule's topology. The core of our generative process is a quantum denoising U-Net (Ronneberger et al., 2015) that learns to reverse the noise process. This architecture is composed of variational quantum circuits (VQCs) for learnable transformations and parameter-free quantum operations for downsampling and upsampling. The training is supervised by structural priors, including node-edge consistency and permutation invariance, to guide the model toward generating valid structures. For efficient inference, we adapt the Denoising Diffusion Implicit Models (DDIMs) (Song et al., 2021a) sampling scheme.

**Our main contributions are summarized as follows:**

1. We propose the first pure quantum diffusion model for 3D de novo molecular generation, featuring a Quantum U-Net architecture that uses VQCs for learnable denoising and parameter-free operators for multi-scale feature processing.

2. We introduce a state preparation method that explicitly encodes atom and bond information from molecular graphs, combined with training supervised by structural priors to guide the generative process.

3. We empirically demonstrate on the QM9 benchmark that our model achieves inference speeds comparable to the fastest classical methods and shows improved generation quality over prominent quantum models, presenting a promising application of QML in scientific discovery.

## 2 RELATED WORK

Our work is situated at the intersection of classical generative models for molecules and the emerging field of QML.

**Classical Generative Models for Molecules.** Deep generative models have significantly advanced molecule discovery. Early approaches were dominated by VAEs (Kingma & Welling, 2013) and GANs (Goodfellow et al., 2014). More recently, diffusion models (Ho et al., 2020) have become the state of the art, particularly for generating 3D molecular structures. Models like Geometric Diffusion Model (GeoDiff) (Xu et al., 2022) and Equivariant Graph Diffusion Model (EDM) (Hoogeboom et al., 2022) demonstrated strong performance by leveraging equivariant neural networks to respect the geometric symmetries of molecules. These models typically operate on continuous representations of atom coordinates and types, progressively denoising them from a simple prior distribution. Key application areas for these models include de novo generation (Hoogeboom et al., 2022), conformer generation (Jing et al., 2022), structure-based drug design (Schneuing et al., 2024), and molecular docking (Corso et al., 2023).

**Accelerating Classical Diffusion Models.** A significant drawback of traditional diffusion models is their slow sampling speed, which requires hundreds or thousands of sequential denoising steps. To address this, DDIMs (Song et al., 2021a) were introduced, enabling faster generation through a deterministic, non-Markovian reverse process. This is achieved by re-formulating the reverse step to depend on the predicted clean data and the current noisy sample, allowing for larger,

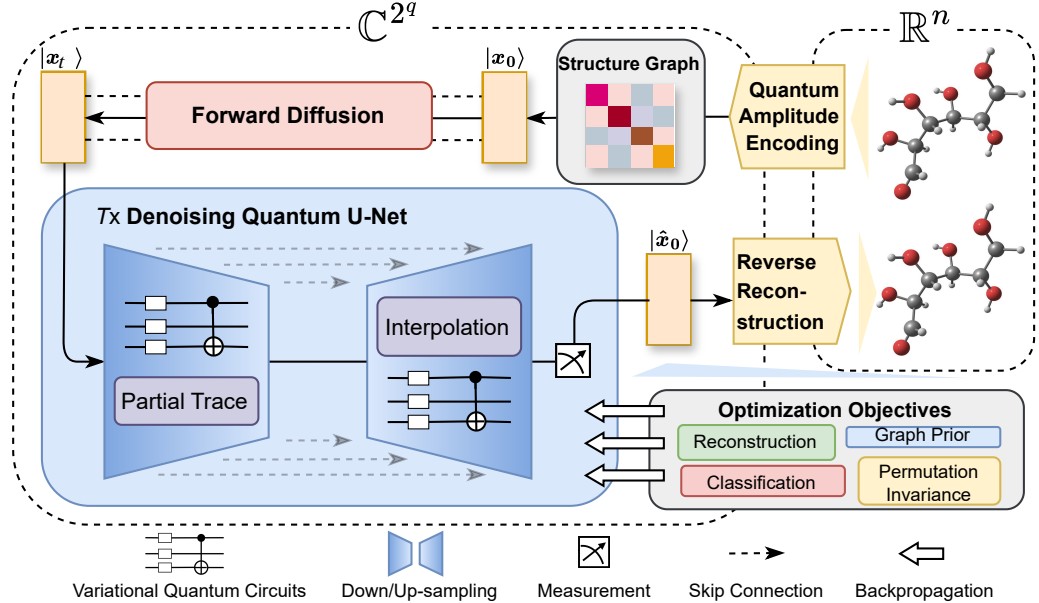

Figure 1: **The workflow of our SQ-Diff model.** A 3D molecular graph is encoded into an initial quantum state $|x_0\rangle$. The forward diffusion process iteratively adds noise, producing a noisy state $|x_T\rangle$. To reverse this, a Quantum U-Net, built from variational quantum circuits and parameter-free down/up-sampling operators, is trained to denoise the state at each timestep. The network is guided by objectives including reconstruction, classification, and graph-based structural priors. During inference, the trained model generates molecular structures from random noise.

more direct jumps along the denoising trajectory. More recently, a new class of models based on flow matching and optimal transport has emerged, offering further acceleration. Geometric Optimal Transport (GOAT) (Hong et al., 2025), for example, formulates molecular graph generation as a time-continuous flow-matching problem, enabling high-quality generation in fewer steps.

**Quantum Generative Models.** QML offers a promising alternative for generative tasks due to the high expressive power of VQCs (Cerezo et al., 2021). Initial forays into molecular generation, such as QGAN-HG (Li et al., 2021) and SQ-VAE (Li & Ghosh, 2022), were primarily hybrid quantum-classical models focused on generating 2D molecular graphs. Subsequent advancements have expanded this landscape with models like MolGAN with VQC (Kao et al., 2023), which explores quantum GANs (Dallaire-Demers & Killoran, 2018) for generative chemistry, hybrid quantum-classical architectures for drug design (Gircha et al., 2023), and quantum-inspired tensor network approaches for small molecular datasets (Moussa et al., 2023). More recent developments include hybrid quantum cycle GANs for small molecule generation (Anoshin et al., 2024), quantum computing-based deep learning for molecular design optimization (Ajagekar & You, 2023), and quantum-enhanced generative models targeting specific inhibitors like KRAS (Ghazi Vakili et al., 2025). The development of pure quantum models for direct 3D generation is more recent and challenging. QVAE-Mole (Wu et al., 2024) was the first to propose a pure quantum VAE for this task. To date, a pure quantum diffusion model for 3D molecular generation has not been proposed, leaving a significant gap that our work aims to fill.

## 3 METHODS

In this section, we present the technical details of our proposed **S**tructure-aware **Q**uantum **Diff**usion (SQ-Diff) model. We first present our novel state preparation method for encoding 3D molecular structures into quantum states. We then describe the pure Quantum U-Net architecture that drives the denoising process. Finally, we introduce the graph-based optimization objectives designed to guide the generation of valid molecules. The workflow is shown in Figure 1. See Appendix A for the quantum basics used in this paper.

## 3.1 QUANTUM STATE PREPARATION FOR MOLECULE GRAPHS

The foundation of our generative process is a novel state preparation procedure that maps a molecule's 3D geometry and atomic features into a quantum state, which is fundamentally different from the classical encoding approach as shown in Figure 2. We begin with the molecule's point cloud data, where each of the $N$ atoms is described by its 3D coordinates $\boldsymbol{a}_i = (a_{ix}, a_{iy}, a_{iz}) \in \mathbb{R}^3$ and a one-hot vector $\boldsymbol{t}_i$ for its atom type. As a preprocessing step, we translate by the per-axis minimum and scale to $[0,1]$: letting $\boldsymbol{a}_{\min} = (\min_N a_{ix}^{ori}, \min_N a_{iy}^{ori}, \min_N a_{iz}^{ori})$ and $D = \max_{1 \le i,j \le N} \|\boldsymbol{a}_i^{ori} - \boldsymbol{a}_j^{ori}\|_\infty$, we set $\boldsymbol{a}_i \leftarrow (\boldsymbol{a}_i^{ori} - \boldsymbol{a}_{\min})/D$. This choice matches the derivation of the unified normalization in Appendix C.1.

**Feature Encoding.** Our method constructs a molecular structure graph $M$ that embeds structural information for the subsequent amplitude encoding. The node-level features, positioned along the diagonal, consist of three components: 1) atom coordinates $\boldsymbol{a}_i$ and 2) atom type one-hot vectors $\boldsymbol{t}_i$, which encode basic node information, and 3) auxiliary values $\sqrt{(2N+1)(3 - \|\boldsymbol{a}_i\|_2^2)}$ inspired by prior work (Wu et al., 2024; Rathi et al., 2023), which facilitate the subsequent normalization. The edge-level features, located in the off-diagonal positions, comprise the Euclidean distance $\|\boldsymbol{a}_i - \boldsymbol{a}_j\|_2$ $(i \ne j)$, which is inherently E(3)-invariant (Garcia Satorras et al., 2021).

**Normalization Scheme.** A key innovation of our work is a state preparation procedure that yields *a unified normalization scheme dependent only on the number of atoms*. We introduce a specific data-dependent correction term $v_{\text{corr}}$ and engineer the feature set such that the squared L2-norm of the flattened feature vector $\boldsymbol{m}$ simplifies to a concise form (see Appendix C.1 for full derivation):

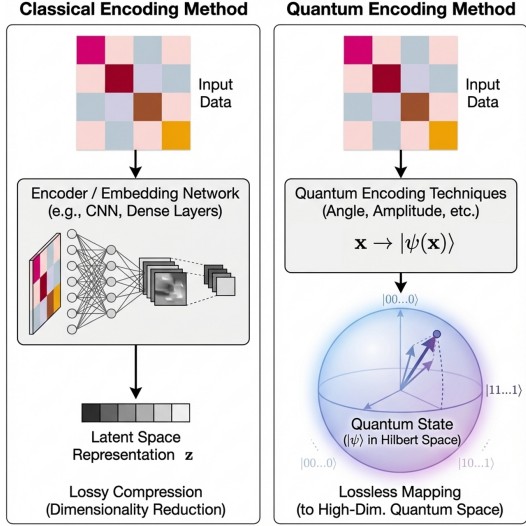

Figure 2: **Comparison of different decoding methods.** The left panel shows the classical decoding method. The right panel shows the quantum decoding method.

$$\|\boldsymbol{m}\|_2^2 = 6N^2 + 4N \quad (1)$$

This result provides a unified normalization paradigm: all entries are normalized by dividing by $\sqrt{6N^2 + 4N}$, which allows the model to process molecules of varying sizes on an equal footing (The form of the initial quantum state is given in Appendix C.1).

The reverse process, **reverse reconstruction**, which maps the final denoised quantum state back to a 3D structure, is the exact symmetric inverse of this preparation step. As it is mathematically straightforward and deterministic, we will not elaborate on it further in a separate section.

## 3.2 QUANTUM FORWARD DIFFUSION PROCESS

The forward diffusion process gradually injects noise into the initial quantum state $|x_0\rangle$ over a sequence of $T$ timesteps. We represent the quantum state $|x_0\rangle$ by its vector of amplitudes, $\boldsymbol{x}_0 \in \mathbb{C}^{2^q}$. The process is defined as a fixed Markov chain that produces a sequence of increasingly noisy latent amplitude vectors $\boldsymbol{x}_1, \ldots, \boldsymbol{x}_T$.

Following standard diffusion models (Ho et al., 2020), the transition at each step $t$ is modeled as a Gaussian kernel with a cosine variance schedule $\{\beta_t \in (0,1)\}_{t=1}^T$:

$$q(\boldsymbol{x}_t | \boldsymbol{x}_{t-1}) = \mathcal{N}(\boldsymbol{x}_t; \sqrt{1 - \beta_t}\boldsymbol{x}_{t-1}, \beta_t \boldsymbol{I}) \quad (2)$$

A notable property of this process is that we can sample $\boldsymbol{x}_t$ at an arbitrary timestep $t$ directly from the initial state $\boldsymbol{x}_0$ in a closed form. Using the notation $\alpha_t = 1 - \beta_t$ and $\bar{\alpha}_t = \prod_{s=1}^t \alpha_s$, the

**Algorithm 1** SQ-Diff Training Procedure
___
1: **Initialize:** Quantum U-Net parameters $\boldsymbol{\theta}$, optimizer.
2: **repeat**
3:     Sample a batch of molecules $\{\boldsymbol{M}_i\}$ from the training dataset $\mathcal{D}$.
4:     **for** each molecule $\boldsymbol{M}$ in the batch **do**
5:         $\boldsymbol{x}_0 \leftarrow \text{StatePreparation}(\boldsymbol{M})$         ▷ Encode molecule into an amplitude vector
6:         $t \sim \text{Uniform}(\{1, \ldots, T\})$                    ▷ Sample a random timestep
7:         $\boldsymbol{\epsilon} \sim \mathcal{N}(\mathbf{0}, \boldsymbol{I})$                          ▷ Sample Gaussian noise
8:         $\boldsymbol{x}_t \leftarrow \sqrt{\bar{\alpha}_t}\boldsymbol{x}_0 + \sqrt{1 - \bar{\alpha}_t}\boldsymbol{\epsilon}$            ▷ Apply forward diffusion process
9:         $\boldsymbol{x}_t \leftarrow \boldsymbol{x}_t / \|\boldsymbol{x}_t\|_2$        ▷ Normalize to get a valid quantum state vector
10:         $|x_t\rangle \leftarrow \text{PrepareStateFromVector}(\boldsymbol{x}_t, t)$         ▷ Embed time and prepare input state
11:         $|\hat{\boldsymbol{x}}_0\rangle \leftarrow \text{QuantumUNet}(|x_t\rangle, t; \boldsymbol{\theta})$         ▷ Predict denoised state with Q-U-Net
12:         $\hat{\boldsymbol{x}}_0 \leftarrow \text{GetAmplitudes}(|\hat{\boldsymbol{x}}_0\rangle)$
13:         Calculate the total loss $\mathcal{L} \leftarrow \mathcal{L}_{\text{recon}} + \lambda_{\text{cls}}\mathcal{L}_{\text{cls}} + \lambda_{\text{prior}}\mathcal{L}_{\text{prior}} + \lambda_{\text{perm}}\mathcal{L}_{\text{perm}}$
14:     **end for**
15:     Update parameters: $\boldsymbol{\theta} \leftarrow \text{OptimizerStep}(\boldsymbol{\theta}, \nabla_{\boldsymbol{\theta}}\mathcal{L})$
16: **until** convergence criteria are met
___

distribution of $\boldsymbol{x}_t$ conditioned on $\boldsymbol{x}_0$ is given by:

$$q(\boldsymbol{x}_t|\boldsymbol{x}_0) = \mathcal{N}(\boldsymbol{x}_t; \sqrt{\bar{\alpha}_t}\boldsymbol{x}_0, (1 - \bar{\alpha}_t)\boldsymbol{I}) \tag{3}$$

This allows for efficient sampling of a noisy vector at any step $t$ via the reparameterization trick:

$$\boldsymbol{x}_t = \sqrt{\bar{\alpha}_t}\boldsymbol{x}_0 + \sqrt{1 - \bar{\alpha}_t}\boldsymbol{\epsilon}, \quad \text{where } \boldsymbol{\epsilon} \sim \mathcal{N}(\mathbf{0}, \boldsymbol{I}) \tag{4}$$

To inform the denoising network of the current noise level, we embed the timestep $t$ using the standard sinusoidal positional encoding (Vaswani et al., 2017), creating a time embedding vector $\boldsymbol{e}_t$. This embedding is added to the noisy state vector $\boldsymbol{x}_t$ after a simple fixed projection to match dimensions. This vector serves as the input amplitude vector for the quantum state $|x_t\rangle$ that is fed into our quantum denoising U-Net.

### 3.3 QUANTUM DENOISING PROCESS

Consistent with the paradigm of classical diffusion models, the reverse process requires a learnable neural network trained to denoise the corrupted state at each timestep. To effectively learn the complex interactions between features at different resolutions, we employ a U-Net-style architecture (Ronneberger et al., 2015), a design empirically validated for its multi-scale analysis capabilities. Our architecture is constructed from a series of VQCs. To ensure the model remains fully within the quantum domain, we introduce parameter-free downsampling and upsampling operators based on partial trace (Romero et al., 2017) and interpolation, respectively, which avoids introducing any classical learnable parameters. During the inference phase, the trained Quantum Denoising U-Net utilizes a DDIM sampling strategy for efficient de novo generation of 3D molecules. Algorithm 1 outlines the end-to-end training process. Algorithm details for generation are provided in Appendix B.

#### 3.3.1 ARCHITECTURE OF THE QUANTUM U-NET

We propose a pure quantum-parameterized denoising U-Net as shown in Figure 3. Analogous to its classical counterpart, our network comprises a downsampling path, an upsampling path, a bottleneck layer, and skip connections that bridge corresponding levels of the two paths. The downsampling path progressively reduces the dimensionality of the quantum state through a series of $n$ blocks. Each block consists of a quantum ansatz $\boldsymbol{U}_{d_i}(\boldsymbol{\theta}_{d_i})$ followed by a downsampling operator $\text{D}_i$. Given an initial input state $|x_t\rangle$, the state at the end of the downsampling path, $|z_t\rangle$, is produced as follows:

$$|z_t\rangle = \left(\prod_{i=1}^{n} \boldsymbol{U}_{d_i}(\boldsymbol{\theta}_{d_i})\text{D}_\text{i}\right)|x_t\rangle \tag{5}$$

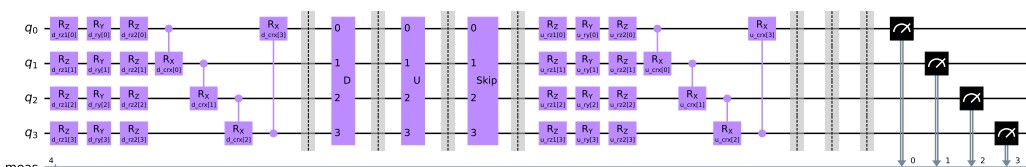

Figure 3: **The structure of one operational block within our Quantum U-Net**, illustrating both the downsampling and upsampling paths. The downsampling path consists of a VQC layer followed by a parameter-free downsampling operator (D). The upsampling path is symmetric, comprising a VQC and an upsampling operator (U), and incorporates information from the corresponding downsampling block via a skip connection.

At the bottleneck, a final quantum ansatz $\boldsymbol{U}_b(\boldsymbol{\theta}_b)$ is applied to this compressed representation to yield the state $|\hat{z}_t\rangle$:

$$|\hat{z}_t\rangle = \boldsymbol{U}_b(\boldsymbol{\theta}_b) |z_t\rangle \qquad (6)$$

The upsampling path is structurally symmetric, iteratively expanding the state representation. A key feature is the incorporation of skip connections, which combine the feature map from the previous upsampling layer with the one from the corresponding downsampling layer. This ensures that high-resolution feature information from early stages is available during the reconstruction process. The final predicted clean state, $|\hat{\boldsymbol{x}}_0\rangle$, is generated by a similar product of upsampling operators $\mathrm{U}_i$ and ansatzes $\boldsymbol{U}_{u_i}(\boldsymbol{\theta}_{u_i})$:

$$|\hat{\boldsymbol{x}}_0\rangle = \left( \prod_{i=1}^{n} \boldsymbol{U}_{u_i}(\boldsymbol{\theta}_{u_i}) \mathrm{U}_i \right) |\hat{z}_t\rangle \qquad (7)$$

Due to the skip connections, the input to the $j$-th upsampling ansatz is a normalized superposition of the output from the previous upsampling block and the state from the corresponding downsampling block. Denoting the output of the $i$-th downsampling block as $|x_t^i\rangle$ and the $j$-th upsampling block as $|\hat{z}_t^j\rangle$, the input is formed as:

$$|\hat{z}_t^j\rangle = \mathrm{Normalize}\left( \boldsymbol{U}_{u_{j-1}}(\boldsymbol{\theta}_{u_{j-1}}) |\hat{z}_t^{j-1}\rangle + |x_t^{n-j+1}\rangle \right) \qquad (8)$$

where $|x_t^0\rangle = |x_t\rangle$ and $|\hat{z}_t^n\rangle = |\hat{\boldsymbol{x}}_0\rangle$. Here, $\mathrm{Normalize}(\cdot)$ denotes L2 renormalization to unit norm. The designs of the quantum ansatz and the training-free down/up-sampling operators are detailed in Appendix C, and the training procedure for the Quantum U-Net is provided in Algorithm 1.

### 3.3.2   DDIM for Efficient Sampling

Standard Denoising Diffusion Probabilistic Model inference requires thousands of sequential denoising steps, a process that is computationally prohibitive for our quantum framework. We therefore adopt the DDIM formalism (Song et al., 2021a) to accelerate sampling, enabling generation in a fraction of the steps with comparable quality. We set $\bar{\alpha}_0 = 1$ by convention.

The DDIM update to obtain $\boldsymbol{x}_{t_{\mathrm{prev}}}$ from $\boldsymbol{x}_t$ can be written as:

$$\boldsymbol{x}_{t_{\mathrm{prev}}} = \sqrt{\bar{\alpha}_{t_{\mathrm{prev}}}} \, \hat{\boldsymbol{x}}_0 \, + \, \sqrt{1 - \bar{\alpha}_{t_{\mathrm{prev}}} - \sigma_t^2} \, \hat{\boldsymbol{\epsilon}}_{\boldsymbol{\theta}} \, + \, \sigma_t \, \boldsymbol{\epsilon}, \qquad (9)$$

where $\boldsymbol{\epsilon} \sim \mathcal{N}(\boldsymbol{0}, \boldsymbol{I})$, $\hat{\boldsymbol{\epsilon}}_{\boldsymbol{\theta}} = (\boldsymbol{x}_t - \sqrt{\bar{\alpha}_t} \, \hat{\boldsymbol{x}}_0)/\sqrt{1 - \bar{\alpha}_t}$, and $\sigma_t = \eta \sqrt{\frac{1 - \bar{\alpha}_{t_{\mathrm{prev}}}}{1 - \bar{\alpha}_t}} \sqrt{1 - \frac{\bar{\alpha}_t}{\bar{\alpha}_{t_{\mathrm{prev}}}}}$. The parameter $\eta$ controls sampling stochasticity. See Algorithm 2 for random generation details.

For conditional generation based on a specific property $y$, we adopt a classifier-guidance strategy (Dhariwal & Nichol, 2021). Given a pre-trained classifier $r_\phi$ that predicts $y$ from a clean-state estimate, we adjust the random prediction by taking a gradient step on a property loss:

$$\hat{\boldsymbol{x}}_0(\boldsymbol{x}_t, t) = \hat{\boldsymbol{x}}_{0,\mathrm{uncond}}(\boldsymbol{x}_t, t) - \lambda \nabla_{\hat{\boldsymbol{x}}_0} \mathcal{L}_{\mathrm{prop}}\big(r_\phi(\hat{\boldsymbol{x}}_{0,\mathrm{uncond}}(\boldsymbol{x}_t, t)), y\big). \qquad (10)$$

The guided $\hat{\boldsymbol{x}}_0$ is then used to compute the corresponding noise for the DDIM update,

$$\hat{\boldsymbol{\epsilon}}_{\boldsymbol{\theta}} = \frac{\boldsymbol{x}_t - \sqrt{\bar{\alpha}_t} \, \hat{\boldsymbol{x}}_0}{\sqrt{1 - \bar{\alpha}_t}}, \qquad (11)$$

which is substituted into the update rule. This framework allows us to generate high-quality molecules in just 10–50 sampling steps. See Algorithm 3 for conditional generation details.

### 3.3.3 STRUCTURE-AWARE OPTIMIZATION OBJECTIVES

Our model is trained by minimizing a composite loss function $\mathcal{L}$ designed to ensure the generation of chemically valid and structurally consistent molecules. The overall objective is a weighted sum of four components:

$$\mathcal{L} = \mathcal{L}_{\text{recon}} + \lambda_{\text{cls}}\mathcal{L}_{\text{cls}} + \lambda_{\text{prior}}\mathcal{L}_{\text{prior}} + \lambda_{\text{perm}}\mathcal{L}_{\text{perm}} \tag{12}$$

The primary component is the reconstruction loss $\mathcal{L}_{\text{recon}}$, which corresponds to predicting the clean state $\boldsymbol{x}_0$:

$$\mathcal{L}_{\text{recon}} = \mathbb{E}_{\boldsymbol{x}_0, \boldsymbol{\epsilon}, t}\left[\left\|\boldsymbol{x}_0 - \hat{\boldsymbol{x}}_0\left(\sqrt{\bar{\alpha}_t}\,\boldsymbol{x}_0 + \sqrt{1 - \bar{\alpha}_t}\,\boldsymbol{\epsilon},\, t\right)\right\|_2^2\right] \tag{13}$$

To enhance the accuracy of atom type prediction, we incorporate a classification loss $\mathcal{L}_{\text{cls}}$ based on the Focal Loss (Lin et al., 2017), which focuses training on hard-to-classify examples. To enforce structural integrity, we introduce a graph prior loss $\mathcal{L}_{\text{prior}}$, which penalizes inconsistencies between reconstructed node features (e.g., coordinates) and edge features (e.g., inter-atomic distances), and ensures that predicted atom-type vectors are valid probability distributions. Finally, because molecular graphs are invariant to node permutations, we include a permutation invariance loss $\mathcal{L}_{\text{perm}}$. This term encourages the model to produce equivalently permuted outputs when the input atoms are re-ordered, thereby embedding this fundamental chemical symmetry into the learning process (Zhou et al., 2020). A detailed formulation of each loss component is provided in Appendix D.

## 4 EXPERIMENTS

In line with Bausch (2020); Ye et al. (2023); Yan et al. (2023); Wu et al. (2024), we conduct our experiments on classical hardware using simulators from the TorchQuantum library (Wang et al., 2022). Our framework is, however, designed to be executable on NISQ computers, leveraging the library's compatibility with interfaces for real quantum devices. Further details on the simulator are provided in Appendix G.

### 4.1 EXPERIMENTAL SETUP

**Datasets.** We evaluate our model on QM9 (Ramakrishnan et al., 2014), a standard benchmark for molecular generation containing about 130K small molecules with up to nine heavy atoms. The dataset provides 3D atomic coordinates and chemical properties. The objective is to train the model for random and conditional generation of molecular structures, including coordinates and atom types (C, N, O, F). We adopt the data partition from Anderson et al. (2019), which designates 100K, 18K, and 10K for training, validation, and testing respectively. Hardware and hyperparameter settings are summarized in Appendix G.

**Baselines.** Our comparative analysis includes two main categories of models. The first consists of **classical methods** with no quantum components, including the continuous normalizing flow model E(n) Equivariant Normalizing Flows (E-NFs) (Garcia Satorras et al., 2021), autoregressive models Generative SchNet (G-SchNet) (Gebauer et al., 2019) and Generative SphereNet (G-SphereNet) (Luo & Ji, 2022), diffusion-based models EDM (Hoogeboom et al., 2022) and Geometric Latent Diffusion Model (GeoLDM) (Xu et al., 2023), and the flow-matching model GOAT (Hong et al., 2025). The second category consists of **hybrid/quantum methods** that utilize trainable quantum parameters. This group includes hybrid models for 2D molecular graph generation, such as SQ-VAE (Li & Ghosh, 2022) and QGAN-HG (Li et al., 2021) (with its variant P2-QGAN-HG), as well as QVAE-Mole (Wu et al., 2024) (with its variant QVAE-Mole (fid.)), a pure quantum VAE for 3D molecular generation.

**Metrics.** We assess model performance based on generation quality and efficiency. To evaluate quality, following prior work (Hoogeboom et al., 2022), we generate 10K molecules and measure their **Validity (V)**, **Uniqueness (U)**, and **Novelty (N)**. Validity is the percentage of molecules adhering to chemical valency rules. Uniqueness is the ratio of distinct molecules among valid samples. Novelty is the proportion of valid molecules not present in the training set. To penalize models that achieve high uniqueness or novelty by generating invalid structures, we report the composite metrics $\mathbf{V} \times \mathbf{U}$,

**V×N**, and **V×U×N**. For efficiency, we report the number of trainable **# C/Q** (classical/quantum) parameters and the effective time **$T_{eff}$**, defined as the time to generate one valid, unique, and novel molecule.

Table 1: **Performance comparison for random 3D molecule generation on the QM9 dataset.** All models generated 10k molecules. The table is divided into classical and quantum-based methods. Quality metrics (V, V×U, V×N, V×U×N): higher is better. Efficiency ($T_{eff}$): lower is better. Best classical score is in red, and best quantum/hybrid score is in blue.

| Methods | Class | V (%) | V×U (%) | V×N (%) | V×U×N (%) | # C/Q | $T_{eff}$ (s) |
|---|---|---|---|---|---|---|---|
| SQ-Diff (Classic) (Subsec. 4.4) | Classic | 70.7 | 3.0 | 70.7 | 3.0 | 0.7M / 0 | 2.69 |
| E-NFs (Garcia Satorras et al., 2021) | Classic | 41.1 | 40.8 | 34.9 | 34.7 | 0.6M / 0 | 0.72 |
| G-SchNet (Gebauer et al., 2019) | Classic | 82.4 | 73.3 | 67.1 | 59.7 | 0.9M / 0 | 0.69 |
| G-SphereNet (Luo & Ji, 2022) | Classic | 82.6 | 29.8 | 37.8 | 13.6 | 3.1M / 0 | 4.04 |
| EDM (Hoogeboom et al., 2022) | Classic | 91.9 | 90.7 | 75.3 | 74.3 | 5.3M / 0 | 1.68 |
| GeoLDM (Xu et al., 2023) | Classic | 93.8 | 92.7 | 54.5 | 53.9 | 12.5M / 0 | 1.86 |
| GOAT (Hong et al., 2025) | Classic | 92.9 | 92.0 | 73.0 | 72.3 | 8.6M / 0 | 0.12 |
| SQ-VAE (Li & Ghosh, 2022) | Hybrid | 44.2 | 7.2 | 16.3 | 2.7 | 128 / 224 | 5.56 |
| QGAN-HG (Li et al., 2021) | Hybrid | 66.6 | 8.1 | 18.5 | 2.2 | 0.5M / 38 | 1.82 |
| P2-QGAN-HG (Li et al., 2021) | Hybrid | 17.6 | 12.4 | 9.5 | 6.7 | 0.1M / 14 | 0.30 |
| QVAE-Mole (Wu et al., 2024) | Quantum | 78.1 | 27.4 | 57.4 | 20.1 | 0 / 602 | 0.40 |
| QVAE-Mole (fid.) (Wu et al., 2024) | Quantum | 74.4 | 26.9 | 31.5 | 11.4 | 0 / 602 | 0.70 |
| **SQ-Diff (Ours)** | Quantum | 80.1 | 48.3 | 79.9 | 48.1 | 0 / 600 | 0.12 |

## 4.2 RANDOM GENERATION

We first evaluate random 3D molecular generation. The results are summarized in Table 1. Our model, SQ-Diff, shows a balance of generation quality and efficiency. Generated examples are visualized in Appendix H.

**Performance Comparison with Classical Methods.** SQ-Diff sets a new standard for pure quantum generative models and demonstrates a promising direction for quantum-inspired discovery. When compared with classical methods, a gap in sample quality remains; for instance, models like GOAT and GeoLDM achieve higher validity. This is an expected outcome, as Quantum Machine Learning (QML) is still an emerging field. The development of quantum models is constrained by the limitations of Noisy Intermediate-Scale Quantum (NISQ) hardware, which restricts the number of qubits, circuit depth, and connectivity, thereby limiting model size and complexity (Bharti et al., 2022). Consequently, it is a known challenge for current QML models, which are often exploratory proofs of concept, to match the performance of highly optimized classical architectures that leverage millions of parameters (Cerezo et al., 2021). As quantum hardware matures to allow for large-scale, fault-tolerant quantum computers, QML models could see significant performance improvements with scale, similar to the scaling law phenomenon in large language models, potentially enabling capabilities not present in smaller-scale models (Kaplan et al., 2020; Wei et al., 2022).

Within this context, the strengths of SQ-Diff are still apparent. A key advantage is its strong capacity for generating novel structures. With a V×N score of 79.9%, our model surpasses all classical baselines in this metric. This may suggest that the quantum model is less prone to posterior collapse and more effectively explores untapped regions of the chemical space. Most importantly from an efficiency standpoint, SQ-Diff matches the inference time ($T_{eff}$) of the state-of-the-art flow-matching method GOAT at 0.12s, while being over 10 times faster than established diffusion models like EDM and GeoLDM. This highlights the potential of quantum-inspired diffusion for high-throughput applications.

**Performance Comparison with Hybrid/Quantum Methods.** Compared to other quantum and hybrid models, SQ-Diff shows a significant improvement in performance. It outperforms existing methods in the composite V×U×N metric with a score of 48.1%, more than double that of the next best pure quantum model, QVAE-Mole (20.1%). This improvement is supported by strong performance across all individual metrics. Specifically, SQ-Diff achieves the highest validity (80.1%) and novelty (79.9%) among all quantum-powered models. Its V×U score of 48.3% indicates greater diversity in the generated samples compared to previous quantum generative models. In terms of

efficiency, SQ-Diff's $T_{eff}$ of 0.12s is on par with the fastest classical methods. It achieves this performance using a comparable number of quantum parameters to QVAE-Mole, making it an effective and efficient pure quantum model for 3D molecular generation.

## 4.3 CONDITIONAL GENERATION

To evaluate the controllability of our model, we performed conditional generation experiments targeting specific chemical properties. Leveraging the classifier-guidance strategy from Section 3.3.2, we aimed to generate molecules with property values around the lower ($Q_{0.25}$) and upper ($Q_{0.75}$) quartiles of the QM9 dataset's distribution for six properties (see Appendix E). This approach serves as a clear test of whether the generation process can be steered toward distinct, pre-defined targets, thereby demonstrating fine-grained control. Figure 4 visualizes the results, displaying the property distributions of 10K molecules generated with guidance toward $Q_{0.25}$ and $Q_{0.75}$, respectively, alongside the distribution from random generation. For all six properties, the guided distributions are clearly shifted and centered around their target quartiles. This demonstrates that our framework provides effective control over the chemical properties of the generated molecules.

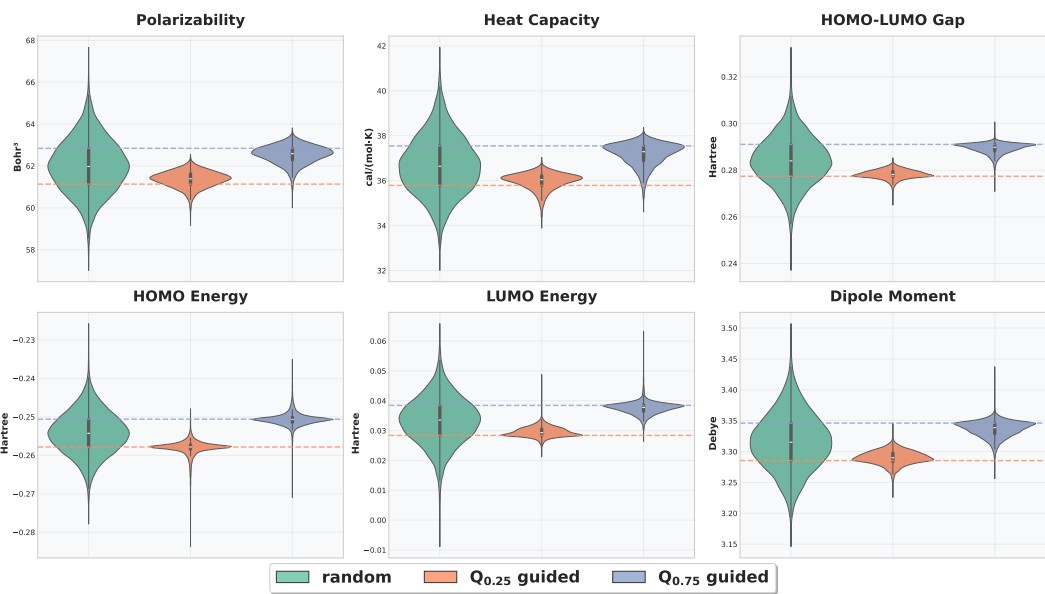

Figure 4: **Distributions of six chemical properties for 10K conditional generated molecules.** For each property, we compare the distribution from random generation (left) with those from conditional generation targeting the lower quartile ($Q_{0.25}$, middle) and upper quartile ($Q_{0.75}$, right).

## 4.4 ABLATION AND SENSITIVITY ANALYSIS

**Impact of Quantum Components.** To verify the contribution of the VQCs, we constructed a classical analogue of SQ-Diff by replacing all learnable VQCs with Multi-Layer Perceptrons (MLPs) of a similar design. As shown in Table 1 (SQ-Diff (Classic)), the results reveal a substantial performance gap. The classical variant shows a drop in generation quality, with the V×U×N metric decreasing from 48.1% to 3.0%. The low uniqueness metric (V×U of 3.0%) suggests that the model experienced mode collapse. Despite hyperparameter tuning, this issue persisted, suggesting that an architecture designed for VQCs does not translate effectively to a classical setting and may be more prone to local minima. This result indicates that the VQCs are more expressive for this task. Furthermore, the classical version is over 20 times slower, with a $T_{eff}$ of 2.69s compared to the quantum model's 0.12s, highlighting the computational advantages of our quantum design.

**Effect of Loss Components.** We performed an ablation study on the loss function components, with results in Table 2. Removing any auxiliary loss term degrades performance. The permutation invariance term ('w/o perm') is the most critical; its removal causes the V×U×N score to drop

Table 2: **Ablation study of the loss function components.**

| Variant | V | V×U | V×N | V×U×N | $T_{eff}$ |
|---|---|---|---|---|---|
| **Full model** | **80.1** | **48.3** | **79.9** | **48.1** | **0.12** |
| w/o cls | 63.0 | 12.3 | 63.0 | 12.3 | 0.49 |
| w/o prior | 40.4 | 14.1 | 40.4 | 14.1 | 0.43 |
| w/o perm | 21.3 | 8.1 | 21.3 | 8.1 | 0.74 |

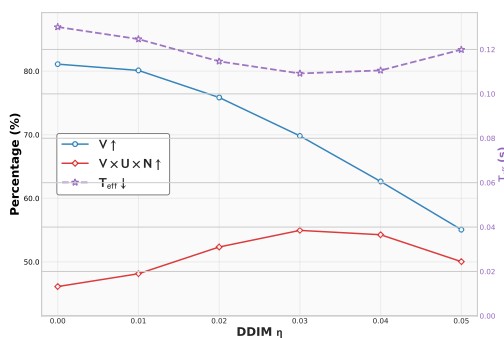

Figure 5: **Sensitivity analysis of the DDIM noise parameter $\eta$.**

to 8.1%, highlighting the importance of encoding chemical symmetry. Removing the graph prior ('w/o prior') or classification ('w/o cls') losses reduces the metric to 14.1% and 12.3%, respectively, confirming their importance for generating valid and diverse molecules.

**Sensitivity to DDIM Noise.** We analyzed the model's sensitivity to the noise parameter ($\eta$) in the DDIM sampler, which controls stochasticity. Figure 5 plots the trade-off between generation quality (V×U×N) and sampling efficiency ($T_{eff}$). The values are provided in Appendix F. Introducing a small amount of noise ($\eta > 0$) improves the V×U×N metric, which peaks around $\eta = 0.03$. This suggests that minor stochasticity helps the model explore the data manifold, enhancing diversity. However, as $\eta$ increases further, performance declines as excess noise degrades chemical validity.

## 5 CONCLUSION

In this work, we introduced SQ-Diff, the first pure quantum diffusion model for de novo 3D molecular generation. Our framework uses a novel state preparation procedure and a pure Quantum U-Net guided by a structure-aware loss function. Experiments on the QM9 dataset show that SQ-Diff generates high-quality molecules, outperforming existing quantum and hybrid models. While a quality gap to classical SOTA remains, its inference efficiency is comparable to the fastest classical methods. SQ-Diff operates with a distinct quantum parameterization, establishing a promising direction for quantum-inspired molecular discovery. For the Limitations and Future work, please see Appendix I.

## ETHICS STATEMENT

The research presented in this paper is foundational and focuses on the development of a novel quantum algorithm, SQ-Diff, for 3D molecule generation. The potential applications of this work lie in accelerating scientific discovery in fields such as drug design and materials science, which are broadly beneficial to society. We have considered the potential societal impacts and do not foresee any direct or immediate negative consequences or ethical risks associated with this work. We are committed to the responsible dissemination of our research and advocate for the ethical use of quantum computing and artificial intelligence technologies.

## REPRODUCIBILITY STATEMENT

To ensure the reproducibility of our results, all necessary components will be made available. The source code for the SQ-Diff model, environment configurations, and scripts to replicate the experiments reported in this paper are released in an anonymous repository at `https://anonymous.4open.science/r/SQ-Diff-0703`, which will be made public upon publication. We have detailed the primary experimental setup in Section 4. Further comprehensive details regarding the model architecture, training hyperparameters, and dataset processing are provided in Appendix B and Appendix G.

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

**Table of Contents**

# A  QUANTUM BASICS

Quantum computing is a computational paradigm based on the principles of quantum mechanics (Nielsen & Chuang, 2010). Unlike classical bits, which can only be in a state of 0 or 1, the fundamental unit of quantum information is the **qubit**.

A qubit can exist in a state of $|0\rangle$, $|1\rangle$, or a linear combination of both, a property known as **superposition**. The state of a single qubit, $|\psi\rangle$, can be written as:

$$|\psi\rangle = \alpha |0\rangle + \beta |1\rangle \tag{14}$$

where $\alpha$ and $\beta$ are complex numbers called probability amplitudes, satisfying the normalization condition $|\alpha|^2 + |\beta|^2 = 1$. The values $|\alpha|^2$ and $|\beta|^2$ represent the probabilities of the qubit collapsing to the state $|0\rangle$ or $|1\rangle$, respectively, upon measurement. A system of $n$ qubits can exist in a superposition of all $2^n$ possible classical states, providing access to an exponentially large computational space.

Quantum computations are performed by applying **quantum gates**, which are unitary transformations that evolve the quantum state. A unitary matrix $U$ has the property that its conjugate transpose $U^\dagger$ is also its inverse, i.e., $U^\dagger U = I$. This property ensures that quantum evolution is reversible and conserves probability. Common single-qubit gates include the Hadamard gate ($H$) for creating superpositions and Pauli gates ($X, Y, Z$). Multi-qubit gates, such as the Controlled-NOT (CNOT) gate, create **entanglement**, a quantum correlation where the state of one qubit is intrinsically linked to another, regardless of the distance separating them (Nielsen & Chuang, 2010).

In hybrid quantum-classical algorithms, a **VQC** is used, comprising a parameterized **ansatz** $U(\theta)$. The ansatz is a circuit template with tunable parameters $\theta$, optimized classically to minimize a cost function and approximate target states or solutions (Nielsen & Chuang, 2010).

Finally, to extract information from a quantum system, a **measurement** is performed. In the standard computational basis ($\{|0\rangle, |1\rangle\}$), measuring the qubit $|\psi\rangle$ causes its superposition to collapse to one of the basis states. The outcome is probabilistic, governed by the amplitudes of the state vector. For a general $n$-qubit state $|\Psi\rangle = \sum_{i=0}^{2^n-1} c_i |i\rangle$, a measurement will yield the outcome corresponding to the basis state $|i\rangle$ with probability $|c_i|^2$.

# B  ALGORITHMS

In this section, we provide the pseudocode for the sampling procedures of our SQ-Diff model. Algorithm 2 details the procedure for random molecular generation using the DDIM sampler. Algorithm 3 extends this to conditional generation using a classifier-guidance strategy.

---

**Algorithm 2** Random Generation

1: **Input:** Trained parameters $\boldsymbol{\theta}$, number of sampling steps $S$, schedule parameter $\eta$.
2: $\boldsymbol{x}_T \sim \mathcal{N}(\boldsymbol{0}, \boldsymbol{I})$         $\triangleright$ Sample initial noise from a standard normal distribution
3: $\boldsymbol{x}_T \leftarrow \boldsymbol{x}_T / \|\boldsymbol{x}_T\|_2$
4: Define timestep sequence $\tau_S, \tau_{S-1}, \ldots, \tau_1$ (from $T$ down to 1).
5: **for** $i = S, \ldots, 1$ **do**
6:      $t \leftarrow \tau_i$, $t_{\text{prev}} \leftarrow \tau_{i-1}$ (where $\tau_0 = 0$)
7:      $|x_t\rangle \leftarrow$ PrepareStateFromVector$(\boldsymbol{x}_t, t)$
8:      $|\hat{\boldsymbol{x}}_0\rangle \leftarrow$ QuantumUNet$(|x_t\rangle, t; \boldsymbol{\theta})$         $\triangleright$ Predict the original state vector
9:      $\hat{\boldsymbol{x}}_0 \leftarrow$ GetAmplitudes$(|\hat{\boldsymbol{x}}_0\rangle)$
10:     $\hat{\boldsymbol{\epsilon}}_{\boldsymbol{\theta}} \leftarrow (\boldsymbol{x}_t - \sqrt{\bar{\alpha}_t}\hat{\boldsymbol{x}}_0)/\sqrt{1 - \bar{\alpha}_t}$         $\triangleright$ Calculate the corresponding noise
11:     $\sigma_t \leftarrow \eta \sqrt{\frac{1 - \bar{\alpha}_{t_{\text{prev}}}}{1 - \bar{\alpha}_t}} \sqrt{1 - \frac{\bar{\alpha}_t}{\bar{\alpha}_{t_{\text{prev}}}}}$         $\triangleright$ Noise schedule
12:     $\boldsymbol{\epsilon} \sim \mathcal{N}(\boldsymbol{0}, \boldsymbol{I})$         $\triangleright$ Sample fresh noise per step
13:     $\boldsymbol{x}_{t_{\text{prev}}} \leftarrow \sqrt{\bar{\alpha}_{t_{\text{prev}}}}\hat{\boldsymbol{x}}_0 + \sqrt{1 - \bar{\alpha}_{t_{\text{prev}}} - \sigma_t^2} \cdot \hat{\boldsymbol{\epsilon}}_{\boldsymbol{\theta}} + \sigma_t \boldsymbol{\epsilon}$         $\triangleright$ DDIM update step
14:     $\boldsymbol{x}_{t_{\text{prev}}} \leftarrow \boldsymbol{x}_{t_{\text{prev}}} / \|\boldsymbol{x}_{t_{\text{prev}}}\|_2$         $\triangleright$ Normalize vector for next step
15: **end for**
16: $M \leftarrow$ ReverseReconstruction$(\boldsymbol{x}_0)$         $\triangleright$ Reconstruct molecule from final vector
17: **return** $M$

---

**Algorithm 3** Conditional Generation

1: **Input:** Trained parameters $\boldsymbol{\theta}$, predictor $r_\phi(\cdot)$, target property $y$, guidance strength $\lambda$, steps $S$, schedule $\eta$.
2: $\boldsymbol{x}_T \sim \mathcal{N}(\boldsymbol{0}, \boldsymbol{I})$ and normalize $\boldsymbol{x}_T$.
3: Define timestep sequence $\tau_S, \tau_{S-1}, \ldots, \tau_1$.
4: **for** $i = S, \ldots, 1$ **do**
5:      $t \leftarrow \tau_i$, $t_{\text{prev}} \leftarrow \tau_{i-1}$
6:      $|x_t\rangle \leftarrow$ PrepareStateFromVector$(\boldsymbol{x}_t, t)$
7:      $|\hat{\boldsymbol{x}}_{0,\text{uncond}}\rangle \leftarrow$ QuantumUNet$(|x_t\rangle, t; \boldsymbol{\theta})$
8:      $\hat{\boldsymbol{x}}_{0,\text{uncond}} \leftarrow$ GetAmplitudes$(|\hat{\boldsymbol{x}}_{0,\text{uncond}}\rangle)$
9:              $\triangleright$ Compute guidance gradient
10:     With torch.enable_grad():
11:        $\boldsymbol{x}_{\text{grad}} \leftarrow \hat{\boldsymbol{x}}_{0,\text{uncond}}.\text{detach}().\text{requires\_grad\_}()$
12:        $\mathcal{L}_{\text{prop}} \leftarrow$ MSE$(r_\phi(\boldsymbol{x}_{\text{grad}}), y)$
13:        $\boldsymbol{g} \leftarrow \nabla_{\boldsymbol{x}_{\text{grad}}} \mathcal{L}_{\text{prop}}$
14:     $\hat{\boldsymbol{x}}_0 \leftarrow \hat{\boldsymbol{x}}_{0,\text{uncond}} - \lambda \cdot \boldsymbol{g}$         $\triangleright$ Apply guidance to the prediction
15:     $\hat{\boldsymbol{x}}_0 \leftarrow \hat{\boldsymbol{x}}_0 / \|\hat{\boldsymbol{x}}_0\|_2$
16:             $\triangleright$ Perform DDIM update using the guided $\hat{\boldsymbol{x}}_0$
17:     $\hat{\boldsymbol{\epsilon}}_{\boldsymbol{\theta}} \leftarrow (\boldsymbol{x}_t - \sqrt{\bar{\alpha}_t}\hat{\boldsymbol{x}}_0)/\sqrt{1 - \bar{\alpha}_t}$
18:     $\sigma_t \leftarrow \eta \sqrt{\frac{1 - \bar{\alpha}_{t_{\text{prev}}}}{1 - \bar{\alpha}_t}} \sqrt{1 - \frac{\bar{\alpha}_t}{\bar{\alpha}_{t_{\text{prev}}}}}$
19:     $\boldsymbol{\epsilon} \sim \mathcal{N}(\boldsymbol{0}, \boldsymbol{I})$
20:     $\boldsymbol{x}_{t_{\text{prev}}} \leftarrow \sqrt{\bar{\alpha}_{t_{\text{prev}}}}\hat{\boldsymbol{x}}_0 + \sqrt{1 - \bar{\alpha}_{t_{\text{prev}}} - \sigma_t^2} \cdot \hat{\boldsymbol{\epsilon}}_{\boldsymbol{\theta}} + \sigma_t \boldsymbol{\epsilon}$
21:     $\boldsymbol{x}_{t_{\text{prev}}} \leftarrow \boldsymbol{x}_{t_{\text{prev}}} / \|\boldsymbol{x}_{t_{\text{prev}}}\|_2$
22: **end for**
23: $M \leftarrow$ ReverseReconstruction$(\boldsymbol{x}_0)$
24: **return** $M$

## C QUANTUM COMPUTING DETAILS

### C.1 DERIVATION OF THE UNIFIED NORMALIZATION TERM

We construct a molecular structure vector $\boldsymbol{m}$ such that its squared L2-norm is independent of the specific molecular geometry, depending only on the number of atoms $N$. This is a crucial step for the amplitude encoding of the initial quantum state (Smaldone et al., 2025). To achieve this, we introduce a specific correction term:

$$v_{\text{corr}} = \frac{\sqrt{2} \cdot N \|\boldsymbol{a}_{\min}\|_2}{\max\limits_{1 \leq i,j \leq N} \|\boldsymbol{a}_i - \boldsymbol{a}_j\|_\infty}, \tag{15}$$

The squared L2-norm of the flattened feature vector $\boldsymbol{m}$ is then calculated as follows:

$$\|\boldsymbol{m}\|_2^2 = \underbrace{\sum_{i=1}^N \|\boldsymbol{a}_i\|_2^2}_{\text{coordinates}} + \underbrace{\sum_{i=1}^N 1}_{\text{atom types}} + \underbrace{(2N+1)\sum_{i=1}^N (3 - \|\boldsymbol{a}_i\|_2^2)}_{\text{auxiliary values}} + \underbrace{\sum_{i,j=1}^N \|\boldsymbol{a}_i - \boldsymbol{a}_j\|_2^2}_{\text{distances}} + \underbrace{v_{\text{corr}}^2}_{\text{correction}}$$

$$= 6N^2 + 4N + \sum_{i=1}^N \|\boldsymbol{a}_i\|_2^2 + 2N\sum_{i=1}^N \|\boldsymbol{a}_i\|_2^2 - 2\sum_{\text{cyc}}(\sum_{i=1}^N a_{i\bullet})^2 - (2N+1)\sum_{i=1}^N \|\boldsymbol{a}_i\|_2^2$$

$$+ v_{\text{corr}}^2$$

$$= 6N^2 + 4N + (1 + 2N - (2N+1))\sum_{i=1}^N \|\boldsymbol{a}_i\|_2^2 - 2\sum_{\text{cyc}}\left(\sum_{i=1}^N \frac{a_{i\bullet}^{ori} - a_{min,\bullet}}{D}\right)^2 + v_{\text{corr}}^2$$

$$= 6N^2 + 4N - \frac{2N^2\|\boldsymbol{a}_{\min}\|_2^2}{D^2} + v_{\text{corr}}^2$$

$$= 6N^2 + 4N, \tag{16}$$

where $(a_{ix}^{ori}, a_{iy}^{ori}, a_{iz}^{ori})$ are the original coordinates of the $i$-th atom before translation, $\boldsymbol{a}_{\min} = (\min_N a_{ix}^{ori}, \min_N a_{iy}^{ori}, \min_N a_{iz}^{ori})$, and $D = \max\limits_{1 \leq i,j \leq N} \|\boldsymbol{a}_i - \boldsymbol{a}_j\|_\infty$. Here, $\sum_{\text{cyc}}(\cdot)$ denotes a cyclic sum over the three coordinate axes $x, y, z$ (the bullet $\bullet$ indicates the axis being summed). This simplification provides a unified normalization term $\sqrt{6N^2 + 4N}$ for all molecules with $N$ atoms. To ensure the feature dimension is exactly $2^q$ for amplitude encoding, we pad the remaining dimensions with 0. The initial quantum state $|x_0\rangle$ is prepared as follows:

$$|x_0\rangle = \frac{1}{\sqrt{6N^2 + 4N}} \sum_{i=1}^N \left( a_{ix}|r_i\rangle + a_{iy}|r_i + 1\rangle + a_{iz}|r_i + 2\rangle + \sum_{j=1}^M \mathbf{1}_{ij}|r_i + 2 + j\rangle \right.$$

$$+ \sum_{k=1, k\neq i}^N \|\boldsymbol{a}_i - \boldsymbol{a}_k\|_2 |r_i + M + 2 + k\rangle + \sqrt{(2N+1)(3 - \|\boldsymbol{a}_i\|_2^2)} |r_i + M + N + 3\rangle \right)$$

$$+ \frac{v_{\text{corr}}}{\sqrt{6N^2 + 4N}} |r_{N+1}\rangle + \sum_{i=N(M+N+3)+2}^{2^q} 0 |r_{N+2} + i\rangle, \tag{17}$$

where $M$ is the number of atom types, $q$ is the number of qubits, and $r_i$ is the index of the $i$-th block.

### C.2 QUANTUM ANSATZ DESIGN

Each learnable block in our Quantum U-Net is a composite of VQCs consisting of ansatzes, as depicted in Figure 3. The design of the ansatz is crucial for the model's expressive power (Cerezo et al., 2021). Our VQC layer is composed of two primary components: a layer of single-qubit rotations and a layer of entangling gates.

The single-qubit rotation layer applies a sequence of three rotations, $R_z$, $R_y$, and another $R_z$, to each qubit independently. This structure is a universal single-qubit gate, meaning it can create any arbitrary single-qubit state (Barenco et al., 1995; Nielsen & Chuang, 2010). For an $n$-qubit system, this layer can be expressed as:

$$\boldsymbol{U}_{\text{rot}}(\boldsymbol{\theta}_{\text{rot}}) = \bigotimes_{i=0}^{n-1} R_z(\theta_{i,2}) R_y(\theta_{i,1}) R_z(\theta_{i,0}) \tag{18}$$

Following the rotation layer, an entangling layer is applied to create correlations between the qubits. We employ a linear entanglement scheme using Controlled-$R_x$ ($CR_x$) gates. A $CR_x(\theta)$ gate applies an $R_x(\theta)$ rotation to a target qubit conditional on the state of a control qubit. This is performed sequentially for adjacent pairs of qubits, a common structure in hardware-efficient ansatzes (Cerezo et al., 2021):

$$\boldsymbol{U}_{\text{ent}}(\boldsymbol{\theta}_{\text{ent}}) = \prod_{i=0}^{n-2} CR_x^{(i,i+1)}(\theta_{i,3}) \tag{19}$$

where $CR_x^{(i,j)}$ denotes a $CR_x$ gate with control qubit $i$ and target qubit $j$. The full VQC layer is the product of these two components, $\boldsymbol{U}(\boldsymbol{\theta}) = \boldsymbol{U}_{\text{ent}}(\boldsymbol{\theta}_{\text{ent}})\boldsymbol{U}_{\text{rot}}(\boldsymbol{\theta}_{\text{rot}})$, where $\boldsymbol{\theta}$ comprises all trainable rotation angles.

### C.3 TRAINING-FREE DOWNSAMPLING AND UPSAMPLING OPERATORS

A core design principle of our framework is to maintain a pure quantum architecture. To this end, we designed downsampling and upsampling operators that are parameter-free and interleave naturally with the unitary operations of the quantum ansatzes.

**Downsampling Operator.** Our downsampling operator is inspired by the partial trace, a technique used in quantum autoencoders to reduce dimensionality (Romero et al., 2017). To downsample an $n$-qubit system to an $m$-qubit system ($m < n$), we partition the Hilbert space into $\mathcal{H} = \mathcal{H}_A \otimes \mathcal{H}_B$, where $\mathcal{H}_A$ has dimension $2^m$ and $\mathcal{H}_B$ has dimension $2^{n-m}$. The state can be written as $|\psi\rangle = \sum_{i=0}^{2^m-1} \sum_{j=0}^{2^{n-m}-1} c_{ij} |i\rangle_A |j\rangle_B$. Our operator projects the ancillary subsystem $B$ onto the basis state $|0\rangle_B$. The resulting unnormalized state of subsystem $A$ is $|\tilde{\psi}_A\rangle = \sum_{i=0}^{2^m-1} c_{i0} |i\rangle_A$. The final downsampled pure state is then obtained by normalization:

$$|\psi_A\rangle = \frac{1}{\sqrt{\sum_{i=0}^{2^m-1} |c_{i0}|^2}} \sum_{i=0}^{2^m-1} c_{i0} |i\rangle_A \tag{20}$$

This procedure effectively reduces the Hilbert space dimensionality while preserving a pure state representation for the subsequent quantum ansatzes.

**Upsampling Operator.** For the upsampling path, we employ a linear interpolation technique, analogous to transpose convolutions in classical U-Nets (Ronneberger et al., 2015). Given an $n$-qubit state $|\psi\rangle_n = \sum_{i=0}^{2^n-1} \alpha_i |i\rangle_n$, we produce an $(n+1)$-qubit state $|\phi\rangle_{n+1}$. We compute an unnormalized amplitude vector $\tilde{\boldsymbol{\beta}}$ of size $2^{n+1}$ by interpolating between adjacent amplitudes in $\boldsymbol{\alpha}$:

$$\tilde{\beta}_{2i} = w_0 \alpha_i + w_1 \alpha_{i+1} \tag{21}$$

$$\tilde{\beta}_{2i+1} = w_1 \alpha_i + w_0 \alpha_{i+1} \tag{22}$$

for $i = 0, \ldots, 2^n - 2$ where $w_0$ and $w_1$ are fixed, non-learnable hyperparameters. This linear transformation can be expressed via a sparse matrix operator $\mathcal{L}$ acting on the amplitude vector. The final state is normalized to have a unit norm:

$$|\phi\rangle_{n+1} = \frac{1}{\|\tilde{\boldsymbol{\beta}}\|_2} \sum_{j=0}^{2^{n+1}-1} \tilde{\beta}_j |j\rangle_{n+1} \tag{23}$$

## D LOSS FUNCTION DETAILS

The total loss function is $\mathcal{L} = \mathcal{L}_{\text{recon}} + \lambda_{\text{cls}}\mathcal{L}_{\text{cls}} + \lambda_{\text{prior}}\mathcal{L}_{\text{prior}} + \lambda_{\text{perm}}\mathcal{L}_{\text{perm}}$.

**Reconstruction Loss.** We use an $\ell_2$ loss on the clean-state prediction:

$$\mathcal{L}_{\text{recon}} = \mathbb{E}_{\boldsymbol{x}_0, \boldsymbol{\epsilon}, t} \left[ \|\boldsymbol{x}_0 - \hat{\boldsymbol{x}}_0(\boldsymbol{x}_t, t)\|_2^2 \right] \tag{24}$$

where $\boldsymbol{x}_t = \sqrt{\bar{\alpha}_t}\, \boldsymbol{x}_0 + \sqrt{1 - \bar{\alpha}_t}\, \boldsymbol{\epsilon}$ and $\hat{\boldsymbol{x}}_0(\cdot)$ denotes the Quantum U-Net prediction of the clean state.

**Classification Loss.** We use Focal Loss (Lin et al., 2017) to improve atom type prediction. Let $\hat{\boldsymbol{y}}_i$ be the predicted probability vector for the type of atom $i$, extracted from the denoised state $\hat{\boldsymbol{x}}_0$. The loss is:

$$\mathcal{L}_{\text{cls}} = -\frac{1}{N} \sum_{i=1}^{N} \sum_{c=1}^{M} y_{ic}(1 - \hat{y}_{ic})^\gamma \log(\hat{y}_{ic}) \tag{25}$$

where $y_{ic}$ is 1 if atom $i$ is of type $c$ and 0 otherwise, and $\gamma$ is a focusing parameter.

**Graph Prior Loss.** This loss enforces internal consistency. It has two parts, $\mathcal{L}_{\text{prior}} = \mathcal{L}_{\text{onehot}} + \mathcal{L}_{\text{dist}}$. The one-hot loss ensures that predicted atom type distributions are valid:

$$\mathcal{L}_{\text{onehot}} = \frac{1}{N} \sum_{i=1}^{N} \left( \sum_{c=1}^{M} \hat{y}_{ic} - 1 \right)^2 \tag{26}$$

The distance consistency loss enforces geometric agreement between reconstructed coordinates $\hat{\boldsymbol{a}}_i$ and reconstructed distances $\hat{d}_{ij}$, a common practice in 3D generative models (Hoogeboom et al., 2022):

$$\mathcal{L}_{\text{dist}} = \frac{1}{N(N-1)} \sum_{i \neq j} \left( \|\hat{\boldsymbol{a}}_i - \hat{\boldsymbol{a}}_j\|_2 - \hat{d}_{ij} \right)^2 \tag{27}$$

**Permutation Invariance Loss.** To enforce permutation invariance, we sample a random permutation $\pi$ and penalize differences between the permuted output and the output of the permuted input. This is a key inductive bias for graph-structured data (Zhou et al., 2020). Let $\hat{\boldsymbol{x}}_0 = f_{\boldsymbol{\theta}}(\boldsymbol{x}_t)$ be the denoising function.

$$\mathcal{L}_{\text{perm}} = \mathbb{E}_{\boldsymbol{x}_t, \pi} \left[ \|\pi(f_{\boldsymbol{\theta}}(\boldsymbol{x}_t)) - f_{\boldsymbol{\theta}}(\pi(\boldsymbol{x}_t))\|_2^2 \right] \tag{28}$$

where $\pi(\cdot)$ is an operator that permutes the atom-related entries in the state vector.

# E   CONDITIONAL GENERATION DETAILS

This section provides details on the chemical properties used for conditional generation, as discussed in Section 4.3. The guidance mechanism relies on a pre-trained classical classifier for each property. We constructed a simple MLP for each of the six properties. The input to each MLP is the flattened molecular feature vector prepared for amplitude encoding, as described in Section 3.1. The six properties are defined as follows:

**Polarizability** ($\alpha$, in Bohr$^3$): A measure of how easily the electron cloud of a molecule can be distorted by an external electric field.

**HOMO Energy** ($\varepsilon_{\text{HOMO}}$, in Hartree): The energy of the highest occupied molecular orbital, related to the molecule's capacity to donate an electron.

**LUMO Energy** ($\varepsilon_{\text{LUMO}}$, in Hartree): The energy of the lowest unoccupied molecular orbital, related to the molecule's capacity to accept an electron.

**HOMO-LUMO Gap** ($\Delta\varepsilon$, in Hartree): The energy difference between HOMO and LUMO, which is an indicator of molecular stability.

**Dipole Moment** ($\mu$, in Debye): A measure of the net molecular polarity, resulting from the separation of positive and negative charges.

**Heat Capacity** ($C_v$, in cal/(mol·K)): The amount of heat required to raise the temperature of one mole of the substance by one degree Kelvin at a constant volume.

Table 3 provides the detailed numerical results for the conditional generation experiments.

Table 3: Comparison of property distributions for 10K molecules generated by SQ-Diff. For the random (baseline) model, $Q_{0.25}$ and $Q_{0.75}$ denote the quartiles of the generated distribution. For the guided model, the columns show the medians of the generated distributions conditioned on the lower ($Q_{0.25}$) and upper ($Q_{0.75}$) quartiles of the target distribution from QM9. RE quantifies the relative error $|\text{median} - \text{target } Q|/|\text{target } Q|$ for each guided condition.

| Property | SQ-Diff (random) | | SQ-Diff (guided) | | RE | |
|---|---|---|---|---|---|---|
| | $Q_{0.25}$ | $Q_{0.75}$ | Median ($Q_{0.25}$) | Median ($Q_{0.75}$) | $Q_{0.25}$ | $Q_{0.75}$ |
| $\alpha$ | 61.13 | 62.84 | 61.39 | 62.59 | 0.419% | 0.409% |
| $\Delta\varepsilon$ | 0.2774 | 0.2910 | 0.2782 | 0.2899 | 0.282% | 0.406% |
| $\varepsilon_{\text{HOMO}}$ | -0.2578 | -0.2506 | -0.2578 | -0.2507 | 0.002% | 0.044% |
| $\varepsilon_{\text{LUMO}}$ | 0.0284 | 0.0385 | 0.0293 | 0.0378 | 3.244% | 1.783% |
| $\mu$ | 3.285 | 3.346 | 3.290 | 3.339 | 0.140% | 0.222% |
| $C_v$ | 35.79 | 37.55 | 36.04 | 37.29 | 0.714% | 0.689% |

## F  SENSITIVITY ANALYSIS DETAILS

Table 4 provides the data for the sensitivity analysis of the DDIM noise ratio hyperparameter ($\eta$), as discussed in Section 4.4.

Table 4: Sensitivity analysis data for DDIM noise ratio $\eta$.

| $\eta$ | V (%) | V×U (%) | V×N (%) | V×U×N (%) | $T_{\text{eff}}$ (s) |
|---|---|---|---|---|---|
| 0.00 | 81.1 | 46.2 | 90.0 | 46.1 | 0.130 |
| 0.01 | 80.1 | 48.3 | 79.9 | 48.1 | 0.125 |
| 0.02 | 75.9 | 52.4 | 75.7 | 52.3 | 0.115 |
| 0.03 | 69.8 | 55.1 | 69.6 | 55.0 | 0.109 |
| 0.04 | 62.7 | 54.5 | 62.4 | 54.3 | 0.111 |
| 0.05 | 55.1 | 50.3 | 54.7 | 50.0 | 0.120 |

## G  EXPERIMENT CONFIGURATION DETAILS

Table 5: Hardware configuration and model hyperparameters used for all experiments.

| **Hardware Configuration** | |
|---|---|
| GPU | NVIDIA GeForce RTX 3090 |
| CPU | Intel Xeon E5-2683 v4 |
| Memory | 128 GB |
| **Model Hyperparameters** | |
| Batch Size | 512 |
| Number of Epochs | 50 |
| Learning Rate | $1 \times 10^{-4}$ |
| Weight Decay | $1 \times 10^{-4}$ |
| Number of Qubits | 8 |
| Number of VQC Blocks | 5 |
| DDIM Sampling Steps | 10 |
| DDIM $\eta$ | 0.01 |

All experiments were conducted using the hardware and hyperparameter settings detailed in Table 5. The selection of hyperparameters was guided by common practices in related QML literature and grid search. Our model is trained using the TorchQuantum (Wang et al., 2022) simulator, which represents VQCs as a series of unitary operations. These operations are executed as standard tensor manipulations within the PyTorch framework. This design choice ensures that the evolution of the quantum state vector is fully differentiable. Consequently, gradients with respect to the VQC parameters can be computed directly via backpropagation, enabling end-to-end training of the model using standard deep learning optimization techniques.

Regarding the coefficients for the loss components ($\lambda_{\text{cls}}$, $\lambda_{\text{prior}}$, $\lambda_{\text{perm}}$) detailed in Section 3.3.3, it is important to note that these are not treated as conventional hyperparameters for tuning. Instead, they serve as static scaling factors. Their primary purpose is to balance the magnitudes of the gradients originating from different loss terms, ensuring that they are on a comparable scale during backpropagation.

## H  MOLECULE VISUALIZATIONS

We provide additional qualitative examples of molecules generated by SQ-Diff; see Figure 6. Note that hydrogen atoms are not explicitly modeled in our method and are added post-hoc using RD-Kit (Landrum et al., 2013) toolkit for visualization.

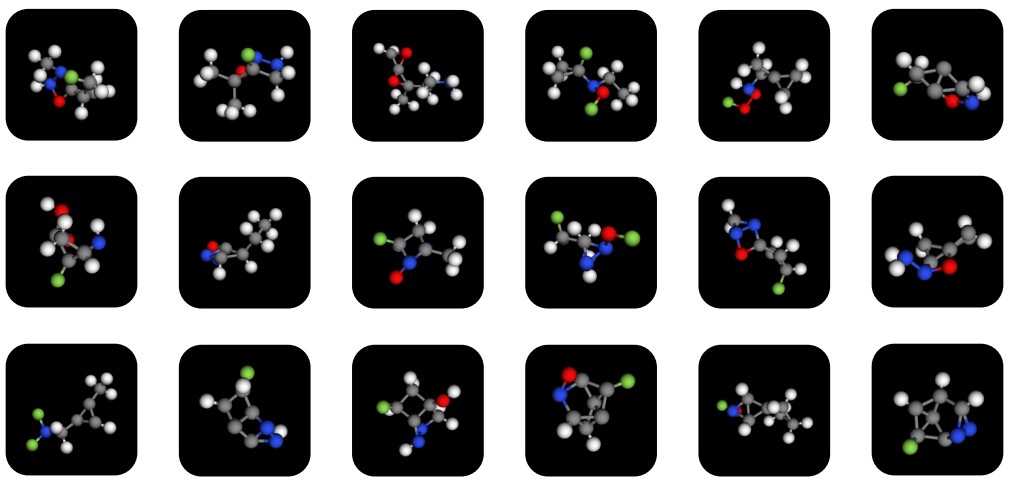

Figure 6: Examples of 3D molecules randomly generated by SQ-Diff.

## I  FUTURE WORK AND LIMITATIONS

Our work represents a significant step towards practical quantum generative modeling for scientific applications, but several limitations and avenues for future research remain. The primary limitation of this study is its reliance on classical simulators for VQCs. While our model is designed with NISQ hardware in mind, its performance on real quantum devices remains to be demonstrated and will be subject to hardware noise and qubit connectivity constraints (Bharti et al., 2022). An immediate and crucial next step is the deployment and validation of SQ-Diff on actual quantum hardware. This will provide invaluable insights into the practical feasibility and potential quantum advantage of our approach. A second limitation is the performance gap in generation quality compared to leading classical models. From a modeling perspective, exploring quantum analogues of more advanced classical generative frameworks, such as flow-matching models (Hong et al., 2025), could lead to further improvements in generation quality and efficiency. Additionally, extending the framework to handle larger and more complex molecular systems, such as proteins, presents an exciting and challenging direction for future work.

## J  THE USE OF LARGE LANGUAGE MODELS

We acknowledge the use of a large language model (LLM) as a writing assistant in the preparation of this manuscript. The LLM was utilized for two primary purposes: (1) to polish the language and improve the clarity, conciseness, and overall expression of the text; and (2) to help identify potential gaps or areas for improvement in the narrative's logical flow. We emphasize that the LLM's role was strictly that of a writing and editing tool. The core scientific ideas, the proposed methodology, the experimental results, and their interpretation are entirely the work of the authors.

