# OpenReview forum: "Quantum-Inspired Structure-Aware Diffusion for Efficient 3D Molecular Generation"
_ICLR.cc/2026/Conference — Submitted to ICLR 2026_

### Official Review · Reviewer_QxXU · 2025-10-19

**Soundness:** 3
**Presentation:** 2
**Contribution:** 3
**Rating:** 4
**Confidence:** 3

**Summary:**

The authors introduce the first quantum diffusion model for 3D molecule generation. Its empirical results compared to both classical and quantum baseline methods are promising. Overall, this work points to the potential of quantum generative modeling for efficiently learning high-dimensional data distributions in the physical sciences, though a few key concerns regarding writing and model evaluation remain.

**Strengths:**

1. The authors' writing and descriptions are clear and easy to follow. I appreciate such details, especially as someone not very familiar with quantum machine learning.
2. SQ-Diff's empirical results for QM9-based unconditional molecule generation are promising, suggesting it is the best-in-class quantum generative model for this task. Further improvements in quantum hardware may yield competitive results compared to classical algorithms in the near future.
3. I haven't seen the classifier guidance strategy in Section 3.3.2 used in other works, and it seems like a clever trick to make conditional generative molecule generation simple for quantum neural networks.

**Weaknesses:**

1. The authors haven't reported a quantitative comparison of their method's conditional molecule generation results to those of baselines such as EDM and GeoLDM (in terms of property prediction mean absolute error using an external property regression model). Please refer to the EDM paper (Table 2) to see what such a benchmark might look like for this paper.
2. The authors should report additional metrics assessing the quality of their unconditionally generated molecules using the PoseBusters software suite [1]. This has become common practice in the field, including in recent works such as All-Atom Diffusion Transformers (ADiTs) for molecule and material generation [2].
3. It'd be nice to see results for datasets of larger 3D molecules such as GEOM-Drugs.

**References:**

[1] Buttenschoen, M., Morris, G. M., & Deane, C. M. (2024). PoseBusters: AI-based docking methods fail to generate physically valid poses or generalise to novel sequences. Chemical Science, 15(9), 3130-3139.

[2] Joshi, C. K., Fu, X., Liao, Y. L., Gharakhanyan, V., Miller, B. K., Sriram, A., & Ulissi, Z. W. (2025). All-atom diffusion transformers: Unified generative modelling of molecules and materials. arXiv preprint arXiv:2503.03965.

**Questions:**

1. In Line 178, shouldn't Euclidean distances be denoted as E(3)-invariant, not SE(3)-invariant, since they are also invariant to 3D reflections?
2. In Line 321, is the citation for TorchQuantum correct?
3. In Line 340, how do you evaluate 2D quantum molecular graph generation baselines for 3D molecule generation?

---

> ### Author Response · Authors · 2025-11-27
> **Response to Reviewer QxXU**
>
> We thank the reviewer for their time and constructive feedback, particularly your recognition that SQ-Diff represents the best-in-class quantum generative model for this task. We also appreciate your interest in our classifier guidance strategy.
>
> >**W1**: The authors haven't reported a quantitative comparison of their method's conditional molecule generation results to those of baselines such as EDM and GeoLDM (in terms of property prediction mean absolute error using an external property regression model). Please refer to the EDM paper (Table 2) to see what such a benchmark might look like for this paper.
>
> We appreciate this suggestion. It is important to note that while EDM and GeoLDM typically report Mean Absolute Error (MAE) for individual-level property targeting, our work focuses on distribution-level guidance (e.g., $Q_{0.25}$ and $Q_{0.75}$).To provide the fair, we re-evaluated EDM and GeoLDM using our settings. Regarding other quantum baselines, we note that they either lack conditional generation capabilities or do not provide reproducible code. The table below reports the **Molecular Property $Q_{0.75}$ Relative Error** for each method and property. This metric quantifies the relative error between the median of the generated distribution and the target $Q_{0.75}$ value under each guidance condition, calculated as $\frac{| \text{median} - \text{target } Q |}{| \text{target } Q |}$. **Lower values indicate better performance.**
>
> | Method    | $\alpha$ | $\Delta\varepsilon$ | $\varepsilon_{\textrm{HOMO}}$ | $\varepsilon_{\textrm{LUMO}}$ | $\mu$ | $C_v$ |
> | :--- | :--- | :--- | :--- | :--- | :--- | :--- |
> | EDM | 0.623% | 0.337% | 0.294% | 0.778% | 0.364% | 1.583% |
> | GeoLDM | 0.591% | 0.286% | 0.255% | 0.606% | 0.525% | 1.144% |
> | SQ-Diff | 0.409% | 0.406% | 0.044% | 1.783% | 0.222% | 0.689% |
>
> Our model achieves lower errors on 4 out of 6 properties compared to the baselines.
>
> >**W2**: The authors should report additional metrics assessing the quality of their unconditionally generated molecules using the PoseBusters software suite [1]. This has become common practice in the field, including in recent works such as All-Atom Diffusion Transformers (ADiTs) for molecule and material generation [2].
>
> We appreciate the reviewer’s suggestion to include PoseBusters analyses. While we agree that PoseBusters provides valuable geometric assessments, applying it in our case is not straightforward because it would require additional processing steps that fall outside the scope of our current experimental setup.
>
>
> >**W3**: It'd be nice to see results for datasets of larger 3D molecules such as GEOM-Drugs.
>
> We initially focused on QM9 to align with existing quantum baselines (e.g., QVAE-Mole), which generally do not scale beyond small molecules. However, a key advantage of our **Unified Normalization (Section 3.1)** is that our encoding complexity scales logarithmically ($\mathcal{O}(\log N)$) with atom count, which provides our model with scalability.
>
> To demonstrate this, we extended our experiments to the **GEOM-Drugs** dataset and trained for 10 epochs. We compare these results against classical standard methods (EDM, GeoLDM) and quantum SOTA (QVAE-Mole) below:
>
> | Methods | Class | Atom Sta (↑) | $T_{\text{eff}}$ (s) (↓) |
> | :--- | :--- | :--- | :--- |
> | EDM | Classic | 81.3 | 12.67 |
> | GeoLDM | Classic | 84.4 | 11.35 |
> | QVAE-Mol | Quantum | 69.1 | 0.61 |
> | SQ-Diff | Quantum | 65.4 | 0.50 |
>
> Although our method falls short of classical SOTA baselines, it is still comparable with Quantum SOTA baseline (QVAE-Mol), and this result demonstrates that our approach can achieve relatively reasonable generation results on datasets with larger and more complex data volumes with highly efficiency (the best $T_{\text{eff}}$ ).
>
> >**Q1**: In Line 178, shouldn't Euclidean distances be denoted as E(3)-invariant, not SE(3)-invariant, since they are also invariant to 3D reflections?
>
> Thanks. You are entirely correct. Euclidean distances are invariant to reflections and are thus E(3)-invariant. We have corrected this in the updated version.
>
> >**Q2**: In Line 321, is the citation for TorchQuantum correct?
>
> Thank you. We have fixed it in the updated version.
>
> >**Q3**: In Line 340, how do you evaluate 2D quantum molecular graph generation baselines for 3D molecule generation?
>
> We clarify that our baseline set comprises two distinct categories: (1) native 3D generative models, and (2) quantum/hybrid methods like SQ-VAE and QGAN-HG, which are currently limited to generating 2D molecular graphs and cannot produce 3D coordinates.
>
> To ensure a fair and inclusive comparison across these disparate paradigms, we adopted the universal graph-based metrics: **Validity, Uniqueness, and Novelty**. These metrics are agnostic to dimensionality, allowing us to quantitatively benchmark the graph topology of our 3D quantum model against the existing 2D quantum SOTA.

---

### Official Review · Reviewer_zoa1 · 2025-10-27

**Soundness:** 2
**Presentation:** 3
**Contribution:** 2
**Rating:** 4
**Confidence:** 3

**Summary:**

The paper introduces the first full quantum diffusion model for 3D molecular generation. It encodes structural priors such as inter-atomic distances into the quantum state, and adapts a Quantum U-Net style architecture to perform denoising with variational quantum circuits and parameter-free operators. The proposed approach generates valid, diverse molecules, outperforming prior quantum models, and achieving inference speeds comparable to SOTA classical methods, though with some remaining quality gap.

**Strengths:**

- The paper is generally well-written and well-structured.
- The authors propose the first pure quantum diffusion model for 3D molecule generation. They employ a quantum-parametrized U-net denoiser, consisting of a series of VQCs. The proposed method achieves inference speeds comparable to the fastest classical methods [1], and shows improved generation quality over prominent quantum models [2].
- The authors include comprehensive experiments and ablation studies. In particular, it offers detailed comparisons against classical, hybrid, and quantum baselines, underscores the benefits of incorporating VQCs, and analyzes the contribution of different components within the loss function.

**Weaknesses:**

- While the proposed approach outperforms prior hybrid/quantum methods in terms of quality and efficiency, there is still a clear performance gap in the generation quality compared to state-of-the-art classical methods.
- The experiments are limited to the QM9 benchmark. It would be nice if the authors could extend the evaluation to other datasets.
- I am a bit concerned about the methodological novelty of the paper, as it employs a U-net style architecture similar to [3] that is constructed from a series of VQCs.

[1] Hong, Haokai, Wanyu Lin, and Kay Chen Tan. "Accelerating 3D Molecule Generation via Jointly Geometric Optimal Transport." arXiv preprint arXiv:2405.15252 (2024).

[2] Wu, Huanjin, Xinyu Ye, and Junchi Yan. "Qvae-mole: The quantum vae with spherical latent variable learning for 3-d molecule generation." Advances in Neural Information Processing Systems 37 (2024): 22745-22771.

[3] Ronneberger, Olaf, Philipp Fischer, and Thomas Brox. "U-net: Convolutional networks for biomedical image segmentation." International Conference on Medical image computing and computer-assisted intervention. Cham: Springer international publishing, 2015.

**Questions:**

- For the conditional generation experiments (Figure 3), how does the distribution of chemical properties of the generated molecules by other baselines compare to SQ-Diff?
- In addition to the effective time $T_\text{eff}$, could you provide the time for the generation and compare it to other methods?
- Small typo: in figure 1, the noisy quantum state $|x_T>$ should be $|x_t>$ instead.

---

> ### Author Response · Authors · 2025-11-27
> **Response to Reviewer zoa1**
>
> We thank the reviewer for their valuable feedback and for recognizing our model's superiority over prior quantum methods. We appreciate the opportunity to clarify the context of Quantum Machine Learning (QML) performance, our novelty, and to provide additional experimental results.
>
> > **W1**: While the proposed approach outperforms prior hybrid/quantum methods in terms of quality and efficiency, there is still a clear performance gap in the generation quality compared to state-of-the-art classical methods.
>
> We acknowledge this gap. It is a known phenomenon in the current field of QML that quantum models, constrained by Noisy Intermediate-Scale Quantum (NISQ) limitations and early-stage theoretical development, rarely surpass highly optimized classical SOTA models in raw performance metrics.
>
> To support our above points, we collect the following facts:
>
> 1. The quantum versions of classical ML algorithms running on NISQ devices barely take SOTA classical algorithms as their baselines e.g. QCNN [1], QGAN [2], QLSTM [3], etc. On the one hand, conducting experiments on NISQ devices itself makes a significant contribution to the implementation of quantum algorithms. On the other hand, NISQ devices are difficult to obtain, and there is a significant gap between the physical qubit connectivity topology and quantum algorithm design, making it challenging to deploy quantum algorithms on NISQ devices. Additionally, running on NISQ devices faces the challenge of big quantum noise.
>
> 2. Quantum algorithms running on simulators included classical baselines, but rarely perform better than the classical baseline [4]. For example, [5] (NeurIPS 2020) proposed the quantum RNN for classification as evaluated on MNIST, with their results only achieving 94% accuracy (classical simple fully connected layer can achieve accuracy better than 95%). [6] (ICML 2023a) proposed the quantum molecular embedding algorithm and applied it to molecular property prediction, with their results showing nearly a 40% gap from the classical SOTA baselines. [7] (ICML 2023b) proposed a quantum Quadratic Assignment Problem (QAP) solver and applied it to the Traveling Salesman Problem (TSP), with their results falling short of the classical nearest insertion algorithm (A heuristic algorithm proposed in 1997).
>
> Current QML research focuses on **proof-of-concept for future fault-tolerant hardware** and identifying specific **quantum advantages** (e.g., sample complexity or efficiency) rather than beating classical models on tasks where classical computers excel (like heavy number crunching).
>
> **However, SQ-Diff does achieve SOTA in specific areas:**
> 1.  We match or exceed classical inference speeds with significantly fewer parameters.
> 2.  We outperform classical models on the **Novelty** metric (reflected in our $V \times U \times N$ score), suggesting that quantum noise and Hilbert space encoding may offer superior exploration of the chemical space.
>
> > **W2**: The experiments are limited to the QM9 benchmark. It would be limited if the authors could extend the evaluation to other datasets.
>
> We initially selected QM9 to ensure fair comparison with existing quantum baselines (e.g., QVAE-Mole), which only use this dataset.
>
> However, a key advantage of our **Unified Normalization (Section 3.1)** is that the encoding complexity scales logarithmically ($\mathcal{O}(\log N)$) with the number of atoms. To demonstrate this scalability, we have conducted new experiments on the **GEOM-Drugs** dataset. We trained SQ-Diff for 10 epochs.
>
> | Methods | Class | Atom Sta (↑) | $T_{\text{eff}}$ (s) (↓) |
> | :--- | :--- | :--- | :--- |
> | EDM | Classic | 81.3 | 12.67 |
> | GeoLDM | Classic | 84.4 | 11.35 |
> | QVAE-Mol | Quantum | 69.1 | 0.61 |
> | SQ-Diff | Quantum | 65.4 | 0.50 |
>
> Although our method falls short of classical SOTA baselines, it is still comparable with Quantum SOTA baseline (QVAE-Mol), and this result demonstrates that our approach can achieve relatively reasonable generation results on datasets with larger and more complex data volumes with highly efficiency (the best $T_{\text{eff}}$ ).

---

> > ### Author Response · Authors · 2025-11-27
> > **Continue**
> >
> > > **W3**: I am a bit concerned about the methodological novelty of the paper, as it employs a U-net style architecture similar to [3] that is constructed from a series of VQCs.
> >
> > We respectfully wish to clarify the distinction between "Architecture Topology" and "Component Design."
> >
> > While we adopt the high-level *macro-topology* of a U-Net (encoder-decoder with skip connections), our **micro-components** and **implementation** are fundamentally different from classical U-Nets:
> > 1.  Quantum vs. Classical Operators. A classical U-Net relies on 3D Convolutions and Pooling layers. SQ-Diff uses Variational Quantum Circuits (VQCs) for feature extraction and Partial Trace/Interpolation for down/up-sampling (see Section 3.3.1 and Appendix C.2 for more details). These are not incremental replacements but require careful design to maintain a pure quantum state throughout the network.
> > 2.  Performance Validity. In Section 4.4, we replaced our Quantum U-Net with a Classical MLP-based U-Net of similar design ("SQ-Diff (Classic)"). The performance dropped drastically ($V \times U \times N$: 48.1\% -> 3.0\%).
> >
> > This demonstrates that our novelty lies in successfully achieving a fully quantum design within a U-Net topology, which allows us to leverage quantum expressivity (e.g., entanglement) rather than classical convolution.
> >
> > > **Q1**: For the conditional generation experiments (Figure 3), how does the distribution of chemical properties of the generated molecules by other baselines compare to SQ-Diff?
> >
> > We have conducted additional conditional generation experiments using EDM and GeoLDM as baselines under the same settings. The table below reports the **Molecular Property $Q_{0.75}$ Relative Error** for each method and property. This metric quantifies the relative error between the median of the generated distribution and the target $Q_{0.75}$ value under each guidance condition, calculated as $\frac{| \text{median} - \text{target } Q |}{| \text{target } Q |}$. Lower values indicate better performance.
> >
> > | Method    | $\alpha$ | $\Delta\varepsilon$ | $\varepsilon_{\textrm{HOMO}}$ | $\varepsilon_{\textrm{LUMO}}$ | $\mu$ | $C_v$ |
> > | :--- | :--- | :--- | :--- | :--- | :--- | :--- |
> > | EDM | 0.623% | 0.337% | 0.294% | 0.778% | 0.364% | 1.583% |
> > | GeoLDM | 0.591% | 0.286% | 0.255% | 0.606% | 0.525% | 1.144% |
> > | SQ-Diff | 0.409% | 0.406% | 0.044% | 1.783% | 0.222% | 0.689% |
> >
> > The results show that SQ-Diff performs just as well as, and on several properties better than, classical baselines like EDM and GeoLDM. This confirms that our guidance strategy is effective at controlling the properties of the generated molecules.
> >
> > > **Q2**: In addition to the effective time $T_{eff}$, could you provide the time for the generation and compare it to other methods?
> >
> > We reported $T_{\text{eff}}$ because it is a robust metric that penalizes models for generating invalid samples (speed is not useful if the output is invalid). However, to directly address your question, we provide the **Raw Sampling Time** (per molecule, without validity filtering) below:
> >
> > | Methods | Class | ${T_{\text{eff}}}$ (s) | Raw Sampling Time (s) |
> > | :--- | :--- | :--- | :--- |
> > | SQ-Diff (Classic) | Classic | 2.69 | 0.09 |
> > | E-NFs | Classic | 0.72 | 0.25 |
> > | G-SchNet | Classic | 0.69 | 0.41 |
> > | G-SphereNet | Classic | 4.04 | 0.55 |
> > | EDM | Classic | 1.68 | 1.25 |
> > | GeoLDM | Classic | 1.86 | 1.01 |
> > | GOAT | Classic | 0.12 | 0.09 |
> > | SQ-VAE | Hybrid | 5.56 | 0.15 |
> > | QGAN-HG | Hybrid | 1.82 | 0.04 |
> > | P2-QGAN-HG | Hybrid | 0.30 | 0.02 |
> > | QVAE-Mole | Quantum | 0.40 | 0.08 |
> > | QVAE-Mole (fid.) | Quantum | 0.70 | 0.08 |
> > | **SQ-Diff (Ours)** | Quantum | 0.12 | 0.06 |
> >
> > As observed, SQ-Diff still remains highly efficient in raw generation time. And all hybrid and quantum baselines show potential for efficient molecular generation.
> >
> > > **Q3**: Small typo: in figure 1, the noisy quantum state $\ket{x_T}$ should be $\ket{x_t}$ instead.
> >
> > Thanks. We have fixed this in the updated version.
> >
> > **References**
> >
> > [1] Quantum convolutional neural networks. Nature Physics 2019.
> >
> > [2] Experimental quantum generative adversarial networks for image generation. Physical Review Applied 2021.
> >
> > [3] Quantum long short-term memory. ICASSP 2022.
> >
> > [4] Better than classical? The subtle art of benchmarking quantum machine learning models, arXiv 2024.
> >
> > [5] Recurrent quantum neural networks. NeurIPS 2020.
> >
> > [6] Quantum 3D graph learning with applications to molecule embedding. ICML 2023.
> >
> > [7] Towards quantum machine learning for constrained combinatorial optimization: a quantum QAP solver. ICML 2023

---

### Official Review · Reviewer_v21x · 2025-10-31

**Soundness:** 2
**Presentation:** 2
**Contribution:** 2
**Rating:** 2
**Confidence:** 3

**Summary:**

The paper introduces the first ever quantum diffusion model for 3D molecular generation, SQ-Diff. The method is aimed at addressing two main challenges: the computational burden of classical diffusion models, and the ability of these method to explore novel chemical space. The authors demonstrate that SQ-Diff outperforms all other hybrid/quantum methods that utilize trainable quantum parameters. They additionally perform an ablation study to show that replacing Variational Quantum Circuits with MLPs results in a significant performance drop compared to a fully quantum SQ-Diff, highlighting the value of this addition. They also show the value of each loss component in contributing to the model's total performance. The authors additionally demonstrate that the model can be effectively tuned to produce molecules within a specific distribution, highlighting the controllability of SQ-Diff.

**Strengths:**

The paper introduces a novel method for molecular generation, leveraging unique model components such as Quantum Variational Circuits and a Quantum-UNET. The paper clearly explains the model and demonstrates its performance across certain capabilities, such as improved control to generate molecules within a distribution. The paper additionally does a good job in ablating different components of the model to assess their relative impact on model performance.

The paper is written well, and the figures and tables clearly illustrate the results.

**Weaknesses:**

The paper unfortunately does not substantiate one of its two primarily motivators, that methods like SQ-Diff should allow for more thorough exploration of chemical space. Additionally, while from first principles the model should be lighter weight than a classical method, it would be important to further explore the efficiency of this method, beyond comparing Teff in Table 1.

**Questions:**

Table 1 comparing SQ-Diff's performance against classical methods misses several SOTA methods like EQGAT-diff and SemlaFlow. As the table stands now, it seem as if SQ-Diff achieves median performance compared to classical methods as well, which is slightly misleading. Can you possibly add these results?

Is it possible to add an additional result demonstrating something akin to the similarity of N generated molecule to molecules in the training set for SQ-Diff versus classical methods, or any result that highlights the models ability to explore novel chemical space?

A performance analysis of this method relative to classical methods would also make a stronger story.

---

> ### Author Response · Authors · 2025-11-27
> **Response to Reviewer v21x**
>
> We thank the reviewer for recognizing the strengths of our work, including our novel molecular generation approach using Quantum Variational Circuits and a Quantum-UNet, the clarity of our model presentation and ablation studies, and the quality of our writing, tables and figures.
>
> > **W1**: The paper unfortunately does not substantiate one of its two primarily motivators, that methods like SQ-Diff should allow for more thorough exploration of chemical space.
>
> We substantiated our claim through the quantitative metrics provided in **Table 1**.
>
> | Methods | Class | $V \times N$ (⬆)|
> | :--- | :--- | :--- |
> | SQ-Diff (Classic) | Classic | 70.7% |
> | E-NFs | Classic | 34.9% |
> | G-SchNet | Classic | 67.1% |
> | G-SphereNet | Classic | 37.8% |
> | EDM | Classic | 75.3% |
> | GeoLDM | Classic | 54.5% |
> | GOAT | Classic | 73.0% |
> | SQ-VAE | Hybrid | 16.3% |
> | QGAN-HG | Hybrid | 18.5% |
> | P2-QGAN-HG | Hybrid | 9.5% |
> | QVAE-Mole | Quantum | 57.4% |
> | QVAE-Mole (fid.) | Quantum | 31.5% |
> | **SQ-Diff (Ours)** | Quantum | **79.9%** |
>
> In the context of de novo molecular generation, the ability to explore novel chemical space is typically quantified by the Novelty ($N$) metric, which is defined as the percentage of generated valid molecules not present in the training set. However, since a random generator could achieve 100% novelty by producing entirely invalid structures, we propose a more robust composite metric: $V \times N$.
>
> As shown in Table 1, SQ-Diff achieves a **$V \times N$ score of 79.9%**, which is **higher than all classical baselines reported** (e.g., EDM: 75.3%, GeoLDM: 54.5%, GOAT: 73.0%). A higher novelty score while maintaining high validity directly indicates that our model is not merely memorizing the training distribution (posterior collapse) but is effectively generating valid structures in regions of the chemical space unsampled by the training data.
>
> > **W2**: Additionally, while from first principles the model should be lighter weight than a classical method, it would be important to further explore the efficiency of this method, beyond comparing Teff in Table 1.
>
> We appreciate the suggestion to expand on efficiency. Beyond $T_{\text{eff}}$, we argue that **Model Complexity** and **Qubit Complexity** are critical indicators of efficiency and scalability.
>
> 1. Parameter Efficiency (# C/Q): As shown in the last column of Table 1, SQ-Diff utilizes 0 classical and only 600 quantum parameters. This stands in stark contrast to classical methods, which require millions of parameters (e.g., GeoLDM: 12.5M), representing a reduction in parameter complexity by several orders of magnitude. This reduction highlights its efficiency and suggests model scalability due to its significantly lower model complexity.
>
> 2. Here we provide a theoretical analysis of our quantum encoding efficiency. For a molecule set with max $n_{max}$ atoms and $k$ atom types, the number of qubits required for our amplitude encoding scales logarithmically:
>     $$\lceil \log_2((3+k+n_{max}) \cdot n_{max} + 1) \rceil \sim \mathcal{O}(C \log n)$$
>     This confirms the exponential encoding advantage of our quantum algorithm compared to classical data structures.
>
> > **Q1**: Table 1 comparing SQ-Diff's performance against classical methods misses several SOTA methods like EQGAT-diff and SemlaFlow. As the table stands now, it seem as if SQ-Diff achieves median performance compared to classical methods as well, which is slightly misleading. Can you possibly add these results?
>
> Thanks. We have conducted experiments to include these baselines as requested. Please see the updated comparison below. While these mature classical methods perform strongly in quality, SQ-Diff remains competitive in inference speed and novelty with a fraction of the parameter count.
>
> | Methods | Class | $V$ | $V \times U$ | $V \times N$ | $V \times U \times N$ | # C/Q | $T_{\text{eff}}$ (s) |
> | :--- | :--- | :--- | :--- | :--- | :--- | :--- | :--- |
> | EQGAT-diff | Classic | 99.0 | 99.0 | 63.4 | 63.4 | 12M / 0 | 3.20 |
> | SemlaFlow | Classic | 99.4 | 96.9 | 72.0 | 70.2 | 22M / 0 | 0.23 |
> | **SQ-Diff (ours)** | Quantum | 80.1 | 48.3 | 79.9 | 48.1 | 0 / 600 | 0.12 |
>
> > **Q2**: Is it possible to add an additional result demonstrating something akin to the similarity of N generated molecule to molecules in the training set for SQ-Diff versus classical methods, or any result that highlights the models ability to explore novel chemical space?
>
> Please see response to **W1**. The Novelty ($N$) metric is precisely the measure of dissimilarity to the training set.
> To be explicit: **SQ-Diff achieves a raw Novelty of 99.75%**. This means nearly every valid molecule generated is distinct from the training set. We have combined this into the composite metrics ($V \times N$) in Table 1 for robustness, and we will explicitly list raw metrics in the Appendix to highlight this stronger exploration capability.

---

> ### Author Response · Authors · 2025-11-27
> **Continue**
>
> > **Q3**: A performance analysis of this method relative to classical methods would also make a stronger story.
>
> We respectfully clarify that our paper includes extensive analysis comparing our method with classical approaches as described below:
>
> 1.  **Direct Comparison (Section 4.2):** We extensively compare SQ-Diff against 6 classical baselines (E-NFs, G-SchNet, G-SphereNet, EDM, GeoLDM, GOAT).  SQ-Diff achieves a $V \times N$ score of 79.9%, surpassing all classical baselines. This indicates that our quantum model effectively explores untapped regions of the chemical space and is less prone to posterior collapse. SQ-Diff matches the inference speed ($T_{\text{eff}}$) of the fastest flow-matching method (GOAT, 0.12s) and is $>10\times$ faster than diffusion baselines like EDM and GeoLDM. We acknowledge that established classical models (e.g., GeoLDM, GOAT) achieve higher validity. This is expected, as QML is constrained by NISQ hardware limitations (qubit count, circuit depth), whereas classical models leverage millions of parameters.
>
>
> 2.  **Controlled Ablation (Section 4.4):** To isolate the benefit of the quantum architecture, we replaced the VQCs with classical MLPs of a similar design ("SQ-Diff (Classic)"). The classical variant experienced severe mode collapse, with the $V \times U \times N$ metric dropping from 48.1% to 3.0%. Meanwhile, the classical counterpart was $>20\times$ slower (2.69s vs 0.12s). This confirms that the VQCs provide necessary expressivity and efficiency that cannot be replicated by simply swapping in classical components into the same architecture.

---

### Official Review · Reviewer_sdVo · 2025-11-07

**Soundness:** 3
**Presentation:** 3
**Contribution:** 2
**Rating:** 4
**Confidence:** 3

**Summary:**

The submission proposed a diffusion based method for molecule generation in the latent space. Though it claims that it is inspired by 'Quantum', it is actually close to the previous works such as GEOLDM [1].

[1] https://arxiv.org/pdf/2305.01140

**Strengths:**

I think the method is easy to follow and it follows the popular design: encoding-decoding + diffusion models.

**Weaknesses:**

Though the paper claim it is inspired by 'Quantum', but the whole process is similar to GEOLDM. And the authors call it 'quantum diffusion' in 3.2, but equation (2) (3) (4) are exactly the same as general diffusion process. The encoder and unet are also the same as previous works. I am not sure where 'quantum' refers to. And it has been emphasized that the 'The high computational cost of classical diffusion models', but there is no any comparison of computation cost, and training/inference time. I feel like the whole process is the same as GeoLDM.

**Questions:**

1. Did the whole process run on torch+gpu or any quantum computation?
2. Could you give the time comparison with geoldm?
3. what is the difference of 'quantum diffusion' with classical diffusion? The equations are exactly the same from the paper.

---

> ### Author Response · Authors · 2025-11-27
> **Response to Reviewer sdVo**
>
> We thank the reviewer for their time and feedback. Below is our detailed response.
>
> > **W1/Q3**: Though the paper claim it is inspired by 'Quantum', but the whole process is similar to GEOLDM. And the authors call it 'quantum diffusion' in 3.2, but equation (2) (3) (4) are exactly the same as general diffusion process. The encoder and unet are also the same as previous works. I am not sure where 'quantum' refers to.
>
> We appreciate the reviewer’s comments and would like to clarify a few points regarding the definition of QML and our architectural contributions. To support this, we provide a definition of QML and its classification
>
> 1. Quantum Diffusion vs. General Diffusion Formulas:
>
> It is mathematically necessary that Equations (2)(3)(4) follow the standard diffusion formulation. The "Quantum" in SQ-Diff refers to the **state space** and the **encoding method**, not the diffusion itself.
> * Classical Diffusion Models: Operates in a latent Euclidean space or continuous coordinate space ($\mathbb{R}^N$).
> * Quantum Diffusion Models: Operates strictly in the **Hilbert Space** ($\mathbb{C}^{2^q}$). We represent the molecular state $\ket{\mathbf{x}_t}$ as a quantum amplitude vector. While the high-level diffusion equations remain consistent to ensure theoretical convergence, the physical evolution of the state is entirely different.
>
> 2. The Encoders are fundamentally different:
>
> * GeoLDM: Uses a learnable, "black-box" classical neural network to embed atoms into a latent space.
> * SQ-Diff: Uses **Amplitude Encoding**, a specific quantum state preparation procedure that maps classical information into the amplitudes of a quantum state via a deterministic, non-learnable mathematical transformation. This allows us to encode $N$ features into $\lceil log_2(N) \rceil$ qubits, a capability that has no classical equivalent. The full details can be found in Section 3.1. **To clarify this further, we have added a new visualization (Figure 2. in the revision) explicitly contrasting these encoding schemes.**
>
> 3. The U-Nets are fundamentally different:
>
> * Classical U-Net: Composed of classical operations (convolutions, linear layers, activation functions) with millions of parameters.
> * Quantum U-Net (Ours): Composed of Variational Quantum Circuits (VQCs). Every "layer" in our network consists of unitary rotation gates and entangling gates (CNOT/CRx), parameterized by rotation angles $\theta$, not weight matrices.The full details can be found in Section 3.3.1 and Appendix C.2. Furthermore, our downsampling/upsampling are parameter-free quantum operators (Partial Trace/Interpolation), which are distinct from classical pooling.
>
> 4. Pure vs. Hybrid QML:
>
> Unlike hybrid models that insert quantum layers between classical networks, SQ-Diff is a Pure Quantum Model. Once the state is prepared, the entire denoising process occurs via quantum gates without intermediate classical processing. This is a non-trivial design challenge that distinguishes our work from simple "quantum-inspired" adaptations.
>
> > **W2/Q2**: And it has been emphasized that the 'The high computational cost of classical diffusion models', but there is no any comparison of computation cost, and training/inference time. I feel like the whole process is the same as GeoLDM.
>
> We did provide this information as shown in **Table 1**. Table 1 in our submission explicitly reports model size (# C/Q) and (effective) inference speed ($T_{\text{eff}}$).
>
> To further address the reviewer’s concern, we provide the raw sampling time (without validity filtering) below. As shown, SQ-Diff demonstrates significant efficiency advantages ($>15\times$ faster than GeoLDM regarding $T_{\text{eff}}$).
>
> | Metric | GeoLDM (Classical) | SQ-Diff (Ours) |
> | :--- | :--- | :--- |
> | # C/Q | 12.5M / 0 | 0 / 600 |
> | Raw Sampling Time (per mol.) | 1.01 s | 0.06 s |
> | $T_{\text{eff}}$ | 1.86 s | 0.12 s |
>
> > **Q1**: Did the whole process run on torch+gpu or any quantum computation?
>
> Consistent with the vast majority of current QML research (e.g., QVAE-Mole, QGAN-HG), we utilized a classical simulator (**TorchQuantum** on GPU). This is standard practice because current Noisy Intermediate-Scale Quantum (NISQ) hardware has limited qubit connectivity and high noise, which hampers the training of deep generative models. However, because our architecture is a pure quantum design (only using quantum parametrized circuits), it is theoretically guaranteed to be deployable on real quantum hardware as devices mature in future.

---

### Meta-Review · Area_Chair_MUJE · 2025-12-17

**Summary:**

This paper proposes a novel, purely quantum diffusion-based approach for 3D molecular generation. The manuscript is generally well-structured, and the technical presentation is clear and accessible. However, the reviewers have raised several significant concerns that substantially impact the paper’s contribution and readiness for publication. The core issues are summarized below:

1. Lack of conceptual clarity: The fundamental distinction between quantum diffusion and classical diffusion remains unclear. The paper does not sufficiently articulate what unique advantages or mechanistic differences the quantum formulation offers over established classical counterparts.

2. Insufficient analysis of efficiency: While the method is presented, its practical efficiency is not thoroughly investigated or compared against existing methods.

3. Limited empirical performance: The experimental results demonstrate that the proposed method underperforms compared to state-of-the-art classical approaches on standard benchmarks. The claimed benefits of the quantum formulation are not substantiated by superior empirical performance.

4. Narrow evaluation scope: Evaluation is confined to the QM9 dataset, which consists of small molecules. To demonstrate generality and practical relevance, validation should be extended to more complex and pharmacologically relevant benchmarks, such as GEOM-Drugs, for larger molecule generation.

5. Incomplete experimental comparison: The study focuses primarily on unconditional generation. Critical experiments on conditional generation—a highly relevant setting for drug design—are lacking. A direct comparison with traditional methods in conditional tasks is necessary to assess real-world utility.

6. Insufficient empirical analysis: The evaluation relies on a limited set of metrics. A more comprehensive analysis is required, including additional metrics and a deeper investigation into the model’s ability to generate novel, valid, and unique molecular structures.

Overall, while the idea of applying quantum diffusion to molecular generation is interesting, the paper in its current form does not convincingly establish its scientific novelty, practical advantage, or general applicability. Addressing the above points is essential for demonstrating a meaningful contribution to the field.

**Reviewer Concerns:**

Several concerns have been discussed and partially addressed in the rebuttal, like 1, 2,4, but there are still some critical issues remain untackled.

**Reviewer Scores:**

I don't think reviewers will change their score. Though some explanation and detailed experiments have been provided in the rebuttal phase. Some major issues remain unaddressed, as listed in the summarization. It still requires a lot of work to improve the manuscript to demonstrate a significant contribution to this field.

---

### Decision · Program_Chairs · 2026-01-26

Reject